# The origin, evolution and functional divergence of HOOKLESS1 in plants

Qi Wang[1,3], Jingyan Sun[1,3], Ran Wang[1,3], Zhenhua Zhang[1,3], Nana Liu[1], Huanhuan Jin[1], Bojian Zhong [1] & Ziqiang Zhu [1,2✉]

Apical hooks are functional innovations only observed in angiosperms, which effectively protect the apical meristems out of damage during plant seedlings penetrating soil covers. Acetyltransferase like protein HOOKLESS1 (HLS1) in *Arabidopsis thaliana* is required for hook formation. However, the origin and evolution of HLS1 in plants are still not solved. Here, we traced the evolution of HLS1 and found that HLS1 originated in embryophytes. Moreover, we found that Arabidopsis HLS1 delayed plant flowering time, in addition to their well-known functions in apical hook development and newly reported roles in thermomorphogenesis. We further revealed that HLS1 interacted with transcription factor CO and repressed the expression of *FT* to delay flowering. Lastly, we compared the functional divergence of HLS1 among eudicot (*A. thaliana*), bryophytes (*Physcomitrium patens* and *Marchantia polymorpha*) and lycophyte (*Selaginella moellendorffii*). Although *HLS1* from these bryophytes and lycophyte partially rescued the thermomorphogenesis defects in *hls1-1* mutants, the apical hook defects and early flowering phenotypes could not be reversed by either *P. patens, M. polymorpha* or *S. moellendorffii* orthologs. These results illustrate that HLS1 proteins from bryophytes or lycophyte are able to modulate thermomorphogenesis phenotypes in *A. thaliana* likely through a conserved gene regulatory network. Our findings shed new light on the understanding of the functional diversity and origin of HLS1, which controls the most attractive innovations in angiosperms.

[1] College of Life Sciences, Nanjing Normal University, Nanjing 210023, China. [2] Key Laboratory of Molecular Design for Plant Cell Factory of Guangdong Higher Education Institutes, Institute of Plant and Food Science, Department of Biology, Southern University of Science and Technology, 518055 Shenzhen, China. [3]These authors contributed equally: Qi Wang, Jingyan Sun, Ran Wang, Zhenhua Zhang. ✉email: zqzhu@njnu.edu.cn

The process of land colonization is a hallmark event during plant evolution, which then causes the rapid radiation of angiosperms (aka. Darwin's abominable mystery). Angiosperms indeed innovate some unique features, which facilitate their adaptations to environmental changes. For example, apical hooks in angiosperm seedlings are elegant evolutionary innovations. Newly germinated angiosperm (for instance *Arabidopsis thaliana*) seedlings form apical hooks to protect their shoot apical meristems during soil penetration. Once the shoot emerges from the soil, the hook opens and the cotyledons expand for photosynthesis. Dark-grown etiolated seedlings also form apical hooks.

Plants sense the depth and texture of the soil as a mechanical signal, which coordinates the biosynthesis of ethylene[1]. Ethylene triggers exaggerated hook formation, a feature of the so-called triple response[2]. Studies in the reference plant *A. thaliana* have uncovered the signaling paradigms underlying hook development. A forward genetics screen for mutants defective in hook formation identified the *hookless1* (*hls1*) mutant in the early 1990s[2,3]. The *hls1-1* mutant (in which the glutamic acid at position 346 was changed to lysine, HLS1$^{E346K}$) does not display any hook formation when grown in the dark, even in the presence of ethylene, but it shows normal ethylene responses (inhibited growth) in the hypocotyl and root[2]. Screening for suppressors in the *hls1-1* mutant background revealed that AUXIN RESPONSE FACTOR 2 (ARF2) is required for HLS1 function[4].

HLS1 promotes hook formation by controlling the asymmetric distribution of auxin between the concave and convex sides of the hypocotyl, leading to differential cell growth[3]. In addition to functioning as a key regulator of differential cell growth, HLS1 is also recognized as a molecular hub that integrates various exogenous and endogenous cues, such as light[4], auxin[4], jasmonate[5,6], gibberellic acid[7], and salicylic acid[8], to modulate hook angle.

Although *HLS1* has been cloned for more than two decades, its biochemical feature is enigmatic. *HLS1* encodes a protein similar to GCN5 acetyltransferase[3] but it is still not clear whether HLS1 is a bona fide acetyltransferase. *HLS1* is required to modulate the histone H3 acetylation (H3Ac) levels at the *WRKY33* and *ABA INSENSITIVE 5* (*ABI5*) loci, as revealed by chromatin-immunoprecipitation-qPCR (ChIP-qPCR) assays. However, recombinant HLS1 proteins purified from *Escherichia coli* did not have acetyltransferase activity[9]. This discrepancy suggests that HLS1 might act indirectly to modulate histone acetylation. A recent study argued that oligomerization of HLS1 is required for its function, and that light triggers the deoligomerization of HLS1 via a direct interaction between HLS1 and the photoreceptor phytochrome B (phyB)[10]. This photo-responsive deoligomerization of HLS1 inactivates HLS1 and results in hook opening. Although this study presented an updated model for illustrating HLS1 function, how the light signal is transmitted from HLS1 to its downstream targets remains a mystery.

On another side, increasing evidence show that HLS1 might be a multi-functional molecule in plants, in addition to its founding role in hook development. HLS1 directly interacts with PHYTOCHROME-INTERACTING FACTOR4 (PIF4) and co-regulates hypocotyl elongation under high ambient temperatures (i.e., thermomorphogenesis)[11,12]. Quadruple mutants lacking *HLS1* and its three homologs display abnormal embryo patterning, dwarf architecture, and floral defects in adult plants[13], suggesting that HLS plays a pivotal role during plant development.

Although HLS1 is necessary for a variety of developmental events, its origin, and evolutionary history are still not solved. We also do not know whether HLS1 functions in any other signaling pathways and do not understand when its function diverges during evolution. In this study, we tried to use phylogenomic approaches to trace the origin of *HLS1* in plants and accidentally found the role of HLS1 in plant flowering time control.

Flowering at the appropriate time ensures optimal plant fitness and reproduction, which belongs to another functional innovation in angiosperms. Environmental cues, endogenous hormone signals, and plant age are dedicatedly integrated to control flowering time. FLOWERING LOCUS T (FT), the long-sought florigen protein, moves from leaf vascular tissue to the apex, where it induces flowering[14,15]. *FT* mRNA is specifically expressed in vascular tissues, and its expression is tightly controlled by multiple transcription factors, including CONSTANS (CO) and CRY2-interacting bHLH1 (CIB1)[16–18]. These transcription factors integrate environmental cues and endogenous hormone signals to precisely activate *FT* expression, thus ensuring that the plant flowers at the correct time. Both CO and CIB1 bind to the promoter of *FT* to induce its transcription. Moreover, *FT* expression peaks at dusk when CO protein abundance and mRNA expression are at their maximum[19,20]. CIB1 was the first blue light-dependent protein shown to interact with the blue light photoreceptor cryptochrome 2 (CRY2)[17]. CIB1 also physically interacts with CO to coordinate *FT* induction and flowering[21].

Here, we first traced the origin of HLS1 and found that HLS1 originated in bryophytes (~480 Ma), but did not exist in chlorophytes or charophytes. Then we reported that HLS1 in *A. thaliana* repressed *FT* expression and delayed plant flowering time. Further investigations showed that HLS1 directly interacted with transcription factor CO and abrogated its transcriptional activity. Lastly, we overexpressed *HLS1* orthologs from either *Physcomitrium patens*, *Marchantia polymorpha*, or *Selaginella moellendorffii* into *hls1-1* mutants and compared their complementation phenotypes. Interestingly, HLS1 orthologs from bryophytes and lycophyte rescued the thermomorphogenesis defects in *hls1-1* mutants, but could not function in apical hook development and flowering. Therefore, our results substantiated the function of HLS1 in plant flowering time control and provided an evolutionary view of the origin and functional divergence of *HLS1*, one of the enigmatic genes in plants.

## Results

**HLS1 originates in embryophytes.** To trace the origin of HLS1 in plants, we used similarity search and phylogenetic analysis to identify the orthologs of *A. thaliana* HLS1 (AtHLS1) in Archaeplastida using both genomic and transcriptomic data (Supplementary Data 1). We found that AtHLS1 homologs with conserved functional domains and motifs existed in land plants (embryophytes), but did not occur in glaucophyte, rhodophytes, chlorophytes and charophytes (Fig. 1 and Fig. S1). Phylogenetic analyses demonstrated that AtHLS1 orthologs duplicated and resulted in multiple paralogous clades in embryophytes, probably due to the several whole genome duplication events (Fig. 1). Tree topology indicated AtHLS1 orthologs were subdivided into three major groups with strong supports (Group A, B and C) (Fig. 1). Group C included AtHLS1 orthologs from bryophytes, lycophyte and ferns, whereas Group A and B only presented in seed plants (Fig. 1). Group B included AtHLS1 orthologs from gymnosperms, while Group A contained AtHLS1 and AtHLS1 orthologs from angiosperms (Fig. 1). Extensive lineage- or clade-specific duplications occurred during the evolution of HLS1 that lead to massive paralogs of HLS1 in land plants, such as the two different groups in ferns and gymnosperms, respectively (Fig. 1), whereas only *HLS1* in *M. polymorpha* and *Ginkgo biloba* presented single-copy. Our results indicated that AtHLS1 orthologs originated in embryophytes and likely duplicated in the ancestors of angiosperms, resulting in two different paralogous groups of HLS1 in flowering plants (Fig. 1). We also compared the conserved motif

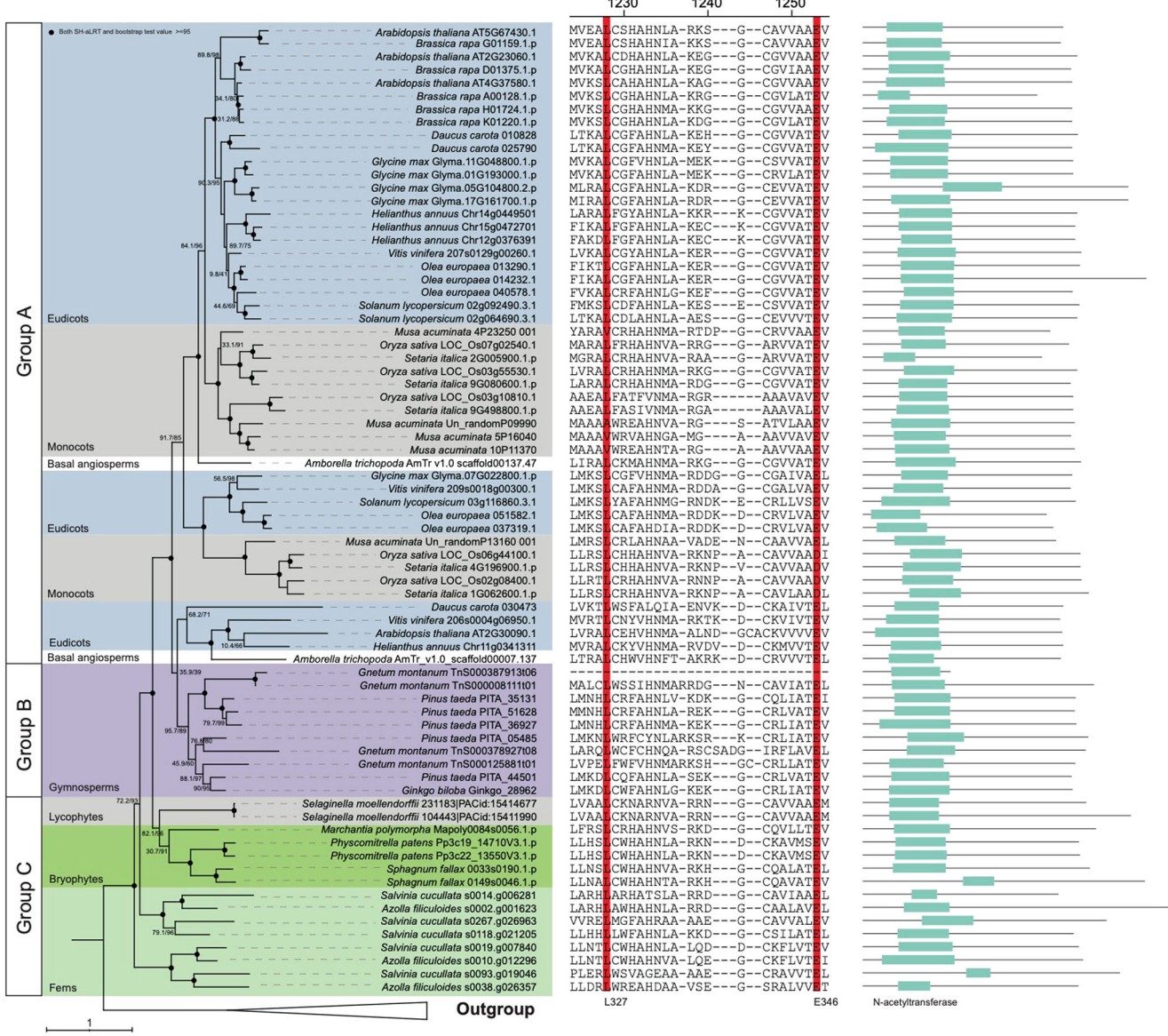

**Fig. 1 Phylogenetic tree and structural comparisons of plant HLS1 orthologs.** Nodal support values are estimated by SH-aLRT test (SH) and ultrafast bootstrap (UFBS) in IQ-TREE2. The "N-acetyltransferase" domain locations were mapped near each branch. The two conserved amino acids in AtHLS1 (L327 and E346) were highlighted in red.

structures and the presence of two conserved residues (L327 and E346 in AtHLS1, as their individual point mutation identified in *hls1-6* (L327W) or *hls1-1* (E346K)) among these HLS1 orthologs. We found that almost all of them contained the five conserved motifs in their N-terminals and two conserved residues in C-terminals, except for one paralog of HLS1 in *Gnetum montanum* (Fig. 1 and Fig. S1). Taken together, our results demonstrated that AtHLS1 orthologs originated in land plants and experienced duplications and divergence in flowering plants, which may correlate with their functional divergence during plant evolution.

**HLS1 delays flowering and represses *FT* expression.** During the propagation of *hls1-1* mutants in the greenhouse, we noticed that these mutants flowered earlier than the wild-type controls. To study how HLS1 might regulate flowering time, we confirmed the early-flowering phenotype in two additional, nonallelic *hls1* mutants (*hls1-27* and *hls1-28*)[12]. All three *hls1* mutant alleles displayed earlier flowering than the wild type under our growth conditions (16 h light/8 h dark, long-day) (Fig. 2). To further

check the relationship between early-flowering and *hls1* mutation, we characterized flowering phenotypes in a genetic complementation line (*35S:MYC-HLS1/hls1-1*). This complementation line successfully rescued the apical hook defects in *hls1-1* mutants (Fig. S2a), while it also reversed the early-flowering phenotypes (Fig. S2b–d). These results illustrate that HLS1 inhibits the initiation of flowering in plants.

Next, we investigated whether HLS1 alters *FT* expression to delay flowering. *FT* is specifically expressed in leaf vascular tissue. Using an *HLS1* promoter-driven glucuronidase (GUS) reporter system (*pHLS1*:GUS), we discovered that *HLS1* was expressed not only in leaf mesophyll cells but also in vascular tissues (Fig. 3a). We then observed GUS activity in the *pHLS1*:GUS plants at different time points of day and found that *pHLS1*:GUS was expressed at higher levels around dusk (Fig. 3b). We also performed quantitative reverse-transcription PCR (qRT-PCR) to further confirm the *HLS1* expression pattern during three consecutive long days. Consistent with the GUS staining results, *HLS1* mRNA levels were elevated during the daytime, peaked at dusk, and decreased at night

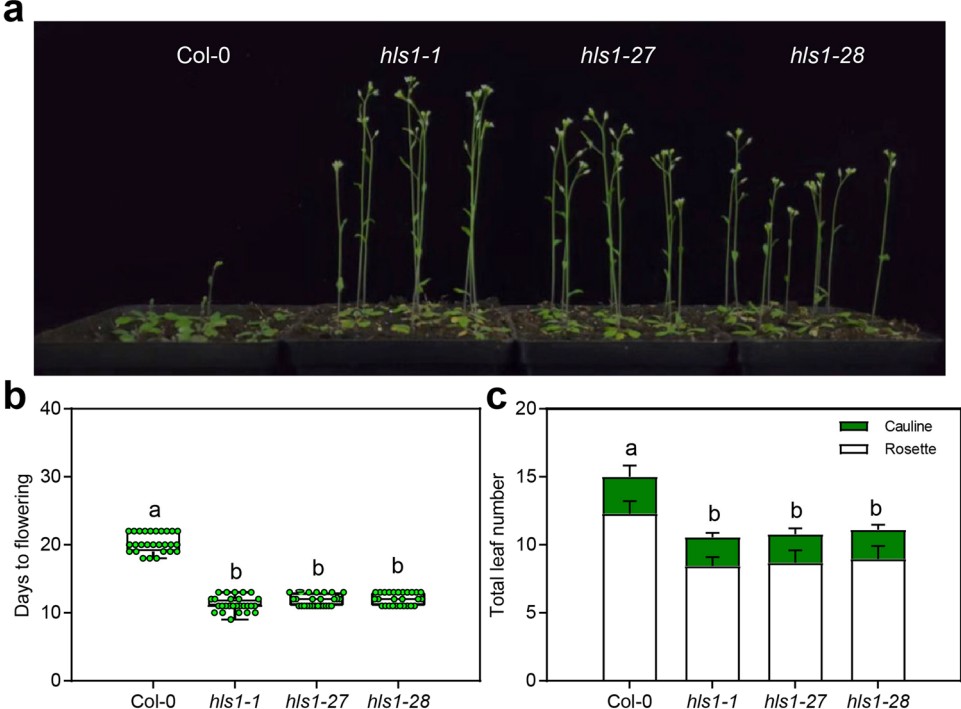

**Fig. 2 HLS1 delays flowering in *Arabidopsis thaliana*. a** Images show representative flowering phenotypes of 22-day-old plants grown under LD conditions (16 h light/8 h dark). **b** Statistical analysis of the days from germination to flowering under LD conditions. Significant differences were determined by one-way analysis of variance (ANOVA) and post hoc Tukey's test, with different lowercase letters indicating significant differences (data are means ± SD; $n = 27$, $P < 0.05$). **c** Statistical analysis of the total leaf number under LD conditions. Significant differences were determined by two-way analysis of variance (ANOVA) and post hoc Tukey's test, with different lowercase letters indicating significant differences (data are means ± SD; $n = 27$, $P < 0.05$).

(Fig. 3c). These spatial and temporal expression patterns of *HLS1* were coincident with the expression profiles of *FT*, suggesting that HLS1 might regulate *FT* transcription. In fact, *FT* expression levels were higher in the *hls1-1* mutants than in the wild type, especially at dusk (Fig. 3d). Taken together, these results demonstrate that HLS1 delays flowering time and represses *FT* expression.

**HLS1 interacts with CO**. We hypothesized that HLS1 suppresses the transcriptional activators of *FT* to repress this gene's expression. To uncover which factor(s) might be involved in this process, we cloned a group of flowering- or light signaling-related transcription factors, and used them as prey proteins to test protein–protein interactions (with HLS1 as a bait) individually in yeast two-hybrid assays. These assays revealed that HLS1 interacted with CO (Fig. 4a). Yeast two-hybrid assays also showed that the carboxyl end of CO (CO dNt: amino acids 133-374) interacted with HLS1 (Fig. 4a). Firefly luciferase complementation imaging (LCI) assays showed that HLS1 interacted with CO *in planta* (Fig. 4b). We also carried out bimolecular fluorescence complementation (BiFC) assays and found that the CO dNt fragment interacted with HLS1 but not the *hls1-1* mutated form of HLS1 (HLS1[E346K]) (Fig. 4c). Taken together, we identified that CO is an HLS1 interacting protein and may contribute to the HLS1-regulated flowering time control.

**HLS1 suppresses the transcriptional activity of CO**. Because the HLS1-interaction domain in CO includes its DNA-binding domain[21], we assumed that HLS1 could compete with CO to repress *FT* expression. We first performed ChIP-qPCR and showed that HLS1 had two major binding regions in the *FT* promoter (region areas were adapted from[22]): one in a remote region (*FT-C* area) and the other in the same site that binds CO (*FT-G* area) (Fig. 5a). Then, we directly tested the regulation of *FT* promoter transcriptional activity. CO-enhanced *luciferase*

(*LUC*) expression driven by the *FT* promoter (529 bp ahead of the start codon, including CO binding sites). However, simultaneously expressing HLS1 with CO dramatically repressed LUC activity (Fig. 5b), confirming that HLS1 represses the transcriptional activity of CO in vivo. To further confirm whether HLS1 represses flowering via CO, we further investigated the genetic relationship between these two genes. The *co* loss-of-function mutants (*co-2*) flowered late under long-day conditions (Fig. S3a), whereas the early-flowering phenotype of *hls1-1* was largely suppressed in the *hls1-1 co-2* double mutants (genotyped in Fig. S3b) (Fig. S3). Because *co-2* allele is in Landsberg (Ler) background, we also crossed *hls1-1* with a *co* mutant in Columbia background (*co-9*). The homozygous line (#21) of *hls1-1 co-9* double mutants (genotyped in Fig. S4) displayed late flowering phenotypes compared with the *hls1-1* parental lines (Fig. 5c–e), similar to the wild-type plants Col-0. These results demonstrate that the role of HLS1 in flowering time control is largely dependent on CO. Furthermore, because *FT* expression levels were elevated in *hls1-1* (Fig. 3d), we characterized the flowering phenotypes of *hls1-1 ft-10* double mutants (genotyped in Fig. S3d) and found that these mutants displayed almost identical flowering phenotypes to the *ft-10* loss-of-function mutants (Fig. S3c). We then quantified the expression levels of *FT* in the parental lines and the *hls1-1 co-9* double mutants. The high expression levels of *FT* in *hls1-1* mutants were largely repressed by the loss of *CO* in the *hls1-1 co-9* double mutants (Fig. 5f). These results indicated that the repression of *FT* expression by HLS1 mainly relied on the presence of CO transcription factor. Taken together, these results demonstrate that HLS1 interacts with CO and occupies the same binding sites in the *FT* promoter as CO to repress *FT* expression.

**Functional divergence of HLS1 orthologs**. Our phylogenomic studies have revealed the origin of HLS1 (Fig. 1), then we are

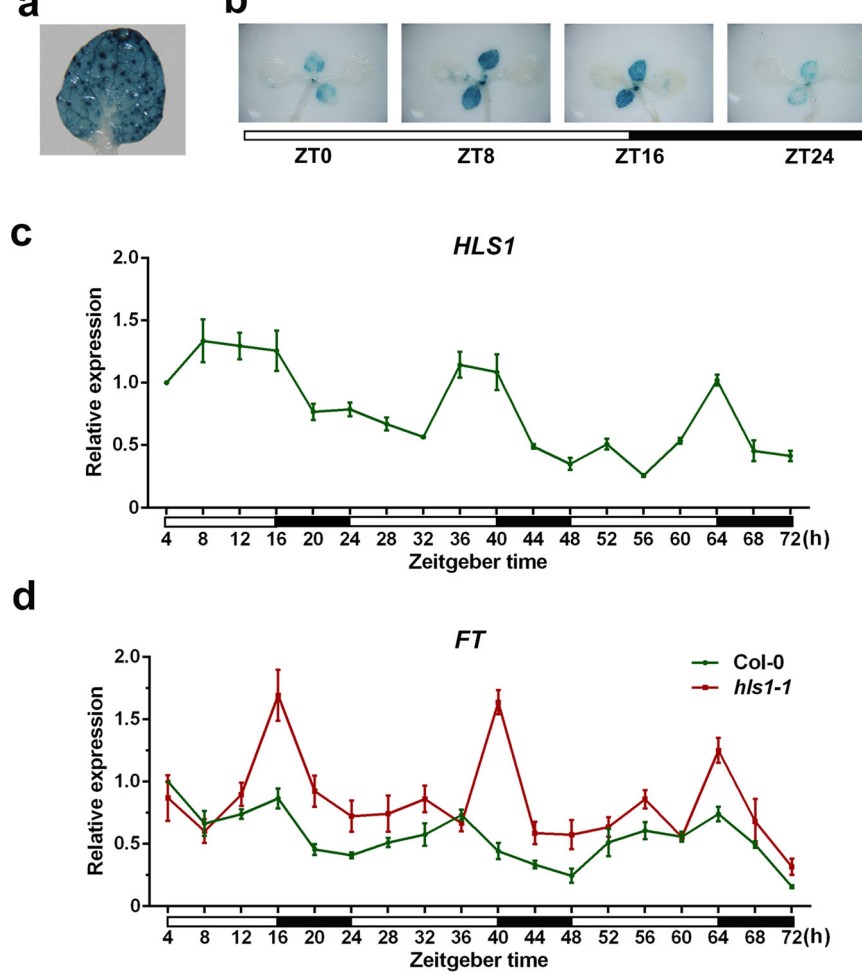

**Fig. 3 HLS1 represses *FT* expression. a** Representative image showing GUS staining in the third leaves of *pHLS1*:GUS plants grown under LD for 12 days. **b** Representative image showing GUS staining at different Zeitgeber times in *pHLS1*:GUS plants grown under LD for 12 days. White bar means daytime and black bar indicates nighttime. **c** qRT-PCR showing *HLS1* expression in 12-day-old Col-0 plants for three consecutive days (LD). White bar means daytime and black bar indicates nighttime. Significant differences were determined by one-way analysis of variance (ANOVA) and post hoc Tukey's test, data are means ± SD; *n* = 3. **d** qRT-PCR showing *FT* expression levels in 12-day-old plants for three consecutive days (LD). White bar means daytime and black bar indicates nighttime. Significant differences were determined by one-way analysis of variance (ANOVA) and post hoc Tukey's test, data are means ± SD; *n* = 3.

wondering about the HLS1 functional divergence during evolution. We selected HLS1 orthologs from bryophytes and lycophyte as ancestral HLS1 for further investigations. In bryophytes, there are one HLS1 ortholog in *M. polymorpha* (liverworts) and two HLS1 orthologs in *P. patens* and *Sphagnum fallax*, respectively (mosses). There are also two HLS1 orthologs in *S. moellendorffii* (lycophyte). Protein sequence alignments in these orthologs and *A. thaliana* AtHLS1 showed that there were two highly conserved regions. One is in their amino terminus, which harbors the conserved acetyltransferase domain (Fig. 1 and Fig. S5), the other is in their carboxyl end. Interestingly, two amino acids (L327 and E346), whose mutation results in hookless phenotype in *A. thaliana*, were strikingly conserved among *A. thaliana*, *M. polymorpha*, *P. patens* and *S. moellendorffii* (Fig. S5). Therefore, we hypothesized that HLS1 orthologs from bryophytes and lycophyte might function in *A. thaliana* to a certain extent.

We initially overexpressed one *HLS1* gene from *M. polymorpha* (*MpHLS1*), two *HLS1* genes from *P. patens* (*PpHLS1-1* and *PpHLS1-2*), and two *HLS1* genes from *S. moellendorffii* (*SmHLS1-1* and *SmHLS1-2*) fused with the LUC tag into *hls1-1* mutants, respectively. With the help of LUC imaging, we easily identified

transgenic lines expressing *LUC-MpHLS1*, *LUC-PpHLS1-1*, *LUC-PpHLS1-2*, *LUC-SmHLS1-1*, or *LUC-SmHLS1-2*, respectively (Fig. S6). Homozygous lines in their propagated generations were used for further studies.

Then we examined three representative phenotypes which are controlled by HLS1. First, we checked thermomorphogenesis responses. HLS1 is required for high ambient temperature responsive hypocotyl elongation[11,12]. In *hls1-1* mutants, their hypocotyls could not elongate under high temperature due to the defects in hypocotyl cell elongation under high temperature[12]. However, overexpression of each *HLS1* ortholog successfully rescued the thermomorphogenesis defects in *hls1-1* mutants (Fig. 6a–d). We also detected the expression levels of *YUCCA8* (*YUC8*), which are up-regulated under high temperature in a HLS1 and PIF4 dependent manner[11]. Our results showed that the induction defects in *hls1-1* mutants were rescued in these complementation lines (Fig. 6e). On another side, we found that no matter *hls1-1* mutants or complementation lines displayed similar root elongation responses under high temperature (Fig. S7), which suggested that HLS1 is not involved in root thermomorphogenesis. It makes sense because HLS1 acts in a

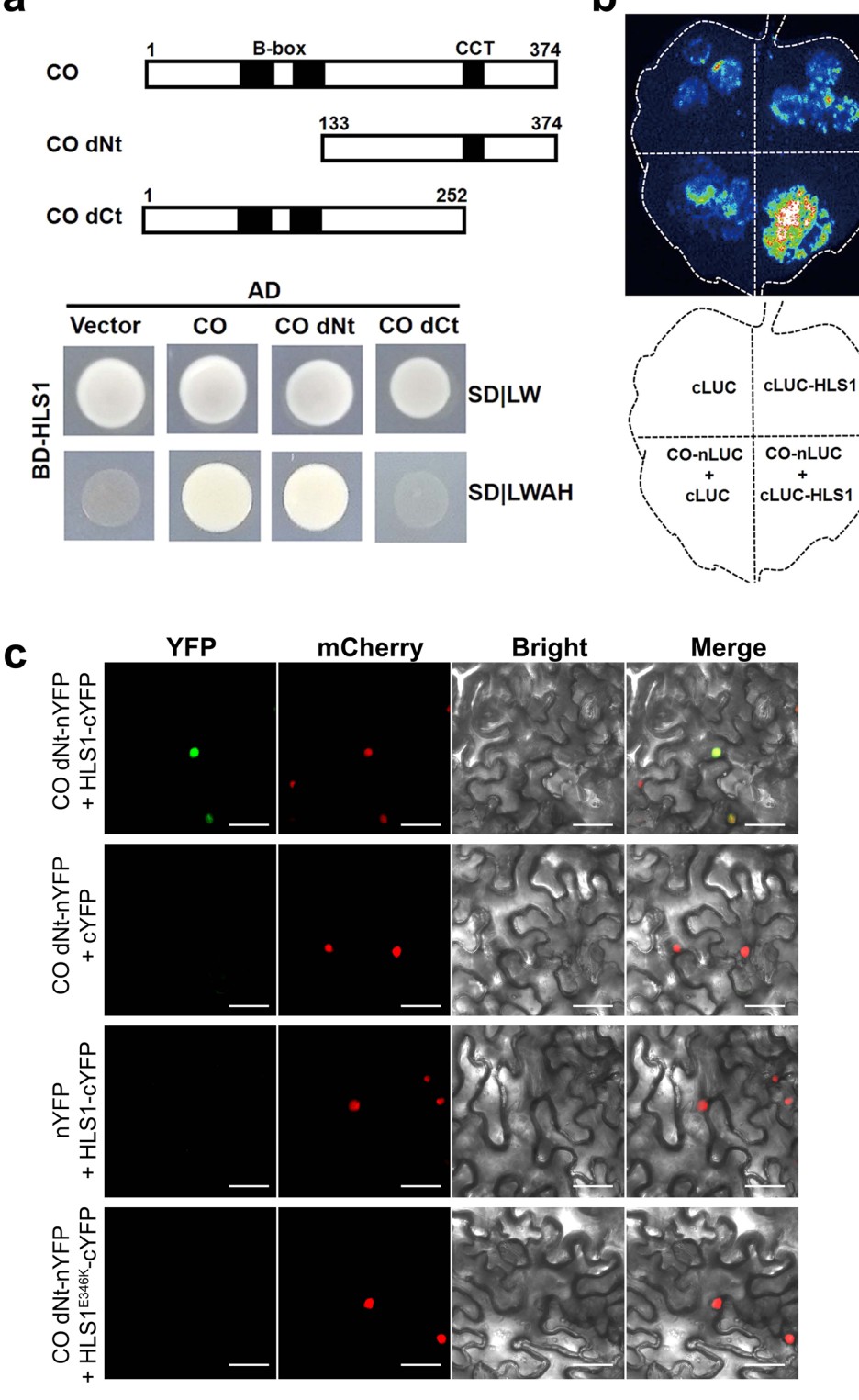

**Fig. 4 HLS1 physically interacts with CO. a** Yeast two-hybrid assays demonstrating that HLS1 interacts with CO. Truncated forms of CO are shown on the top. Numbers indicate the amino acid positions. **b** LCI assay showing that HLS interacts with CO in *N. benthamiana* leaves. *A. tumefaciens* cells harboring each construct were infiltrated into different areas of *N. benthamiana* leaves. After 2–4 days of infiltration, luciferin was sprayed onto the leaf surfaces and LUC activity was recorded. **c** BiFC assay showing HLS1-CO interactions. *A. tumefaciens* cells containing the indicated plasmids were co-infiltrated into *N. benthamiana* leaves for 2 days. YFP fluorescence was monitored under a confocal laser-scanning microscope. The YFP signal (green) channel and the nuclear marker VirD2NLS-mCherry signal (red) channel were merged with bright field images. Bars = 20 μm. nYFP, N-terminal fragment of YFP; cYFP, C-terminal fragment of YFP.

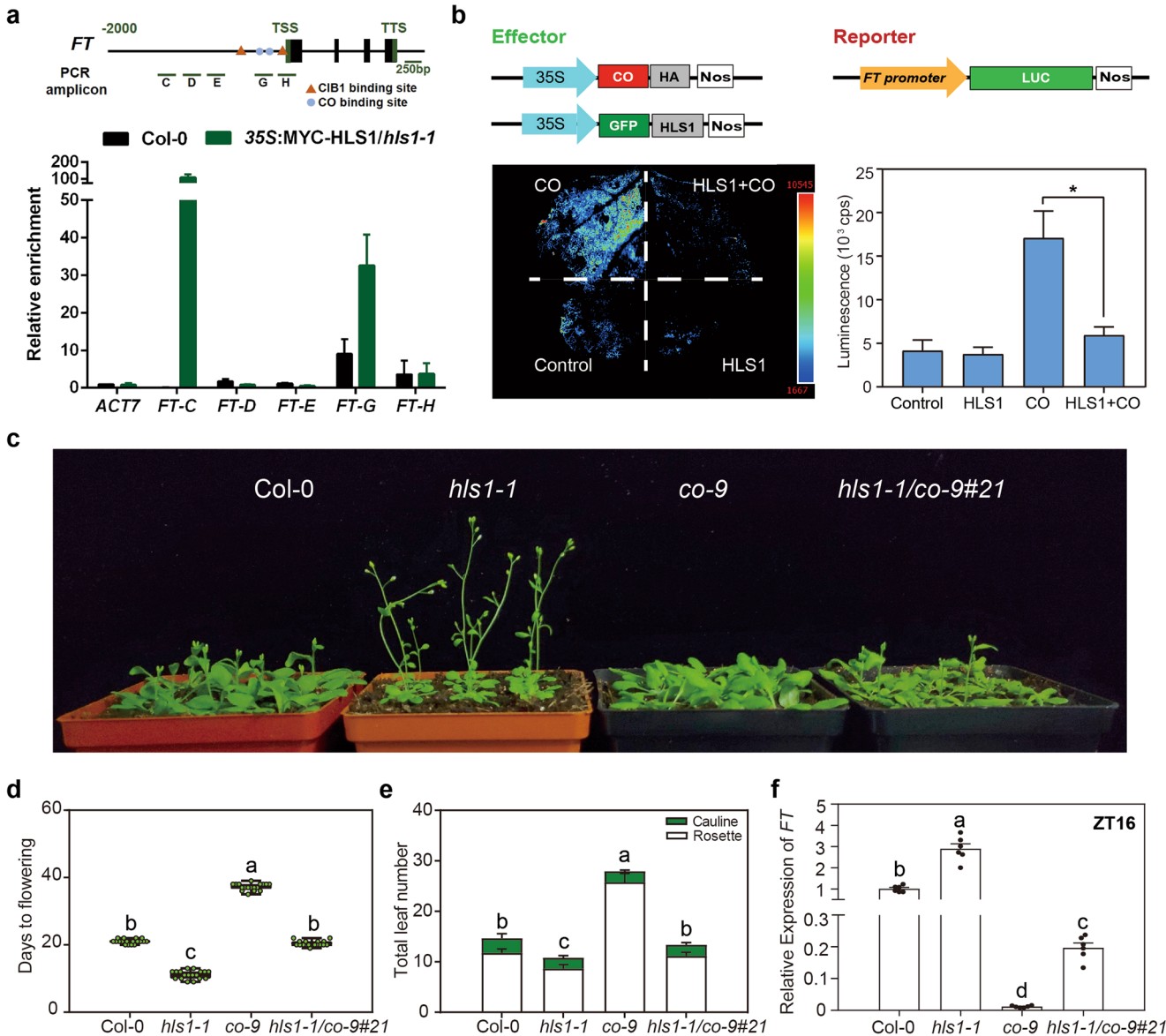

**Fig. 5 HLS1 abrogates CO transcriptional activity. a** ChIP-PCR assays examining the DNA-binding ability of HLS1. Cross-linked chromatin extracted from *35S*:MYC-HLS1/*hls1-1* plants was precipitated with anti-MYC antibody, and the eluted DNA was used to amplify different areas of the *FT* promoter by qPCR. Col-0 plants were used as negative controls. Values shown are means ± SD; *n* = 3. TSS: transcription start site; TTS: transcription terminal site. **b** Transient transcriptional regulation of *FT* promoter activity. *A. tumefaciens* cells harboring *pFT*:LUC (reporter) were co-infiltrated with different effectors into *N. benthamiana* leaves. LUC activity was detected at 3 days after infiltration. Representative infiltrated *N. benthamiana* leaf image detected under CCD camera were shown on the left side. Quantitative results (right side) showing luminescence signal counts per second (cps) in each infiltrated region. Significant differences were determined by one-way analysis of variance (ANOVA) and post hoc Tukey's test, data are means ± SD; *n* = 3, *\*P* < 0.05. **c** Representative image showing 24-day-old plants grown under LD conditions. **d** Statistical analysis of the days from germination to flowering under LD conditions. Significant differences were determined by one-way analysis of variance (ANOVA) and post hoc Tukey's test, with different lowercase letters indicating significant differences (Data are means ± SD; *n* = 18, *P* < 0.05). **e** Statistical analysis of the total leaf number under LD conditions. Significant differences were determined by two-way analysis of variance (ANOVA) and post hoc Tukey's test, with different lowercase letters indicating significant differences (Data are means ± SD; *n* = 18, *P* < 0.05). **f** Expression levels of *FT* in 7-day-old plants grown under LD conditions at ZT16. Significant differences were determined by one-way analysis of variance (ANOVA) and post hoc Tukey's test, with different lowercase letters indicating significant differences (Data are means ± SD; *n* = 6, *P* < 0.05).

PIF4 dependent manner[11], while PIF4 is not involved in root thermomorphogenesis according to several reports[23,24]. Therefore, we concluded that HLS1 from *M. polymorpha*, *P. patens* or *S. moellendorffii* indeed functioned well in the regulation of cell elongation during plant thermomorphogenesis (Fig. 9). Second, we looked into their flowering time. As we demonstrated, *hls1-1* mutants exhibited early-flowering phenotypes. Overexpression of *HLS1* orthologs from bryophytes or lycophyte in *hls1-1* mutants

still flowered early, which meant that these orthologs could not complement *hls1-1* mutants in flowering time control (Fig. 7). Lastly, we observed their hook phenotypes. In etiolated seedlings, plants form closed cotyledons and bended apical hooks (Fig. 8a). Strong mutant alleles of *hls1* (such as *hls1-1*) did not display any hook angles and also exhibited opened cotyledons. However, none of these heterologous overexpression lines displayed any hook angles (Fig. 8b), in contrast to the results from *A. thaliana*

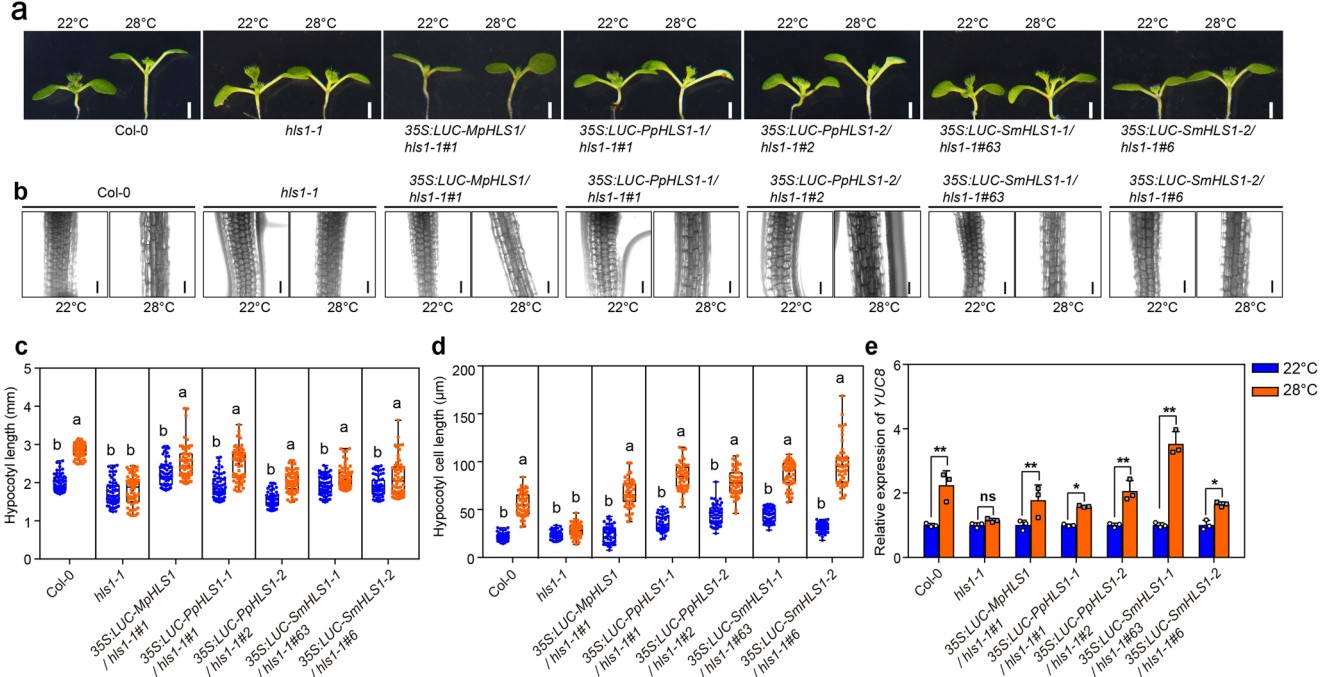

**Fig. 6 Thermomorphogenesis phenotypes in complementation lines. a** Representative images showing the hypocotyl phenotypes in seedlings grown under 22 °C or 28 °C. Scale bar = 1 mm. **b** Representative images showing the hypocotyl cells in the seedlings grown under 22 °C or 28 °C. Bars = 100 μm. **c** Quantification of hypocotyl length. Significant differences were determined by two-way analysis of variance (ANOVA) and post hoc Tukey's test, with different lowercase letters indicating significant differences (Data are means ± SD; $n = 60$, $P < 0.05$). **d** Quantification of hypocotyl cell length. Significant differences were determined by two-way analysis of variance (ANOVA) and post hoc Tukey's test, with different lowercase letters indicating significant differences (Data are means ± SD; $n = 50$, $P < 0.05$). **e** Expression levels of *YUC8* in the seedlings grown under 22 °C or 28 °C. Significant differences were determined by two-way analysis of variance (ANOVA) and post hoc Tukey's test (Data are means ± SD; $n = 3$, *$P < 0.05$, **$P < 0.01$).

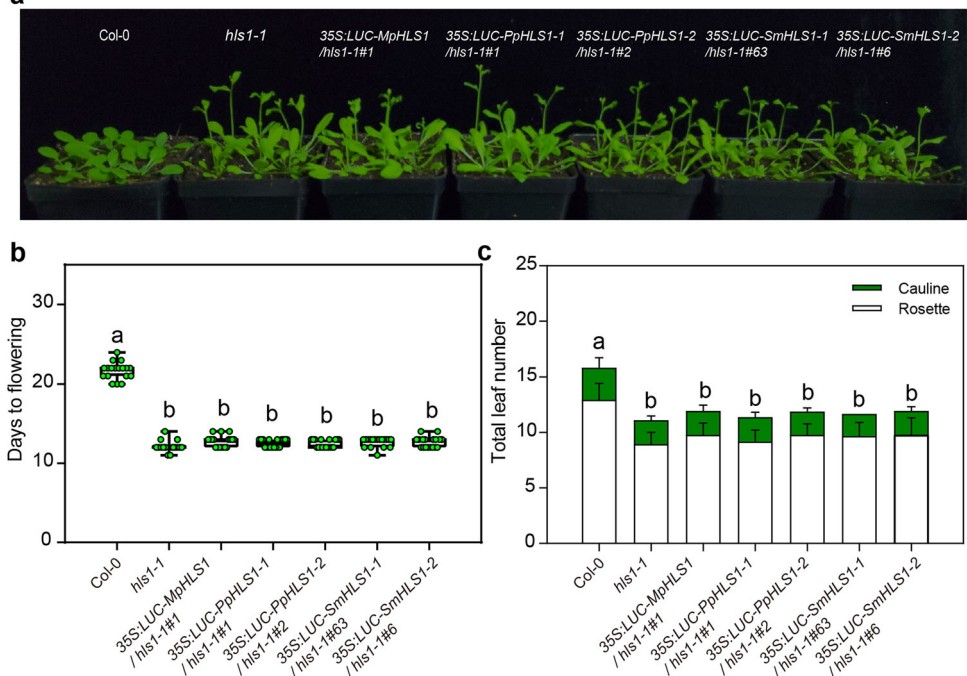

**Fig. 7 Flowering phenotypes under long-day conditions. a** Images show representative flowering phenotypes of 16-day-old plants grown under LD conditions. **b** Statistical analysis of the days from germination to flowering under LD conditions. Significant differences were determined by one-way analysis of variance (ANOVA) and post hoc Tukey's test, with different lowercase letters indicating significant differences (data are means ± SD; $n = 18$, $P < 0.05$). **c** Statistical analysis of the total leaf number under LD conditions. Significant differences were determined by two-way analysis of variance (ANOVA) and post hoc Tukey's test, with different lowercase letters indicating significant differences (data are means ± SD; $n = 18$, $P < 0.05$).

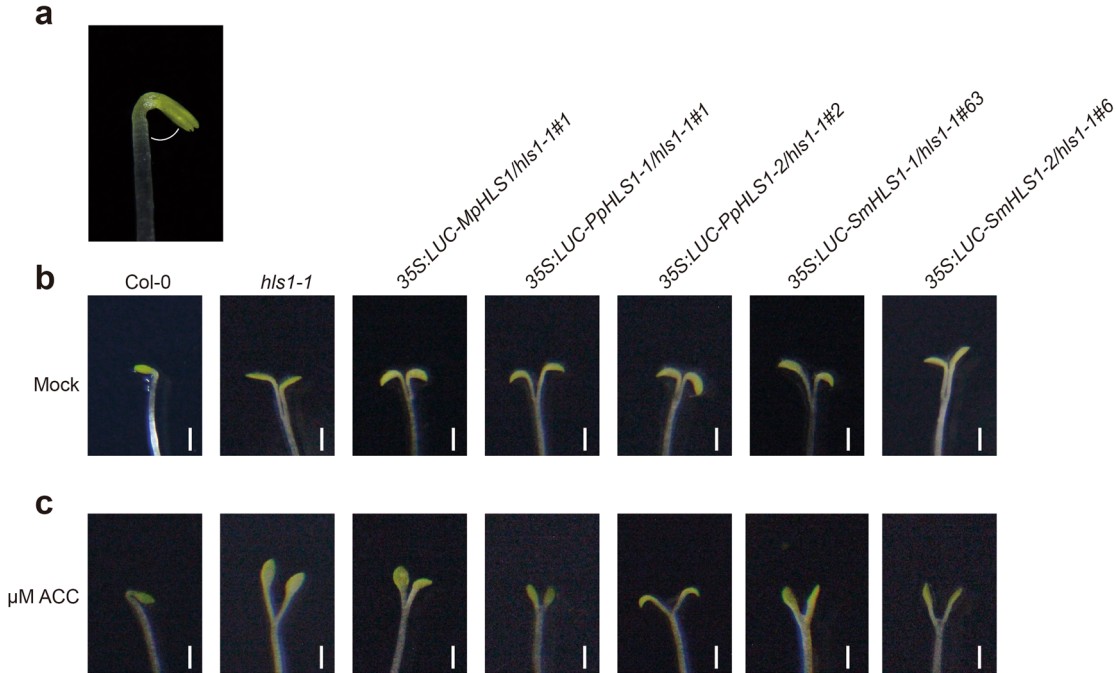

**Fig. 8 Hook angle phenotypes in complementation lines. a** Illustration of the hook angle position in 4-day-old etiolated Col-0 seedling grown on MS medium. **b** Representative images showing hook angles in 4-day-old etiolated seedlings grown on MS medium. **c** Representative images showing hook angles in 4-day-old etiolated seedlings grown on MS medium supplemented with 1 μM of ACC. Scale bars = 1 mm.

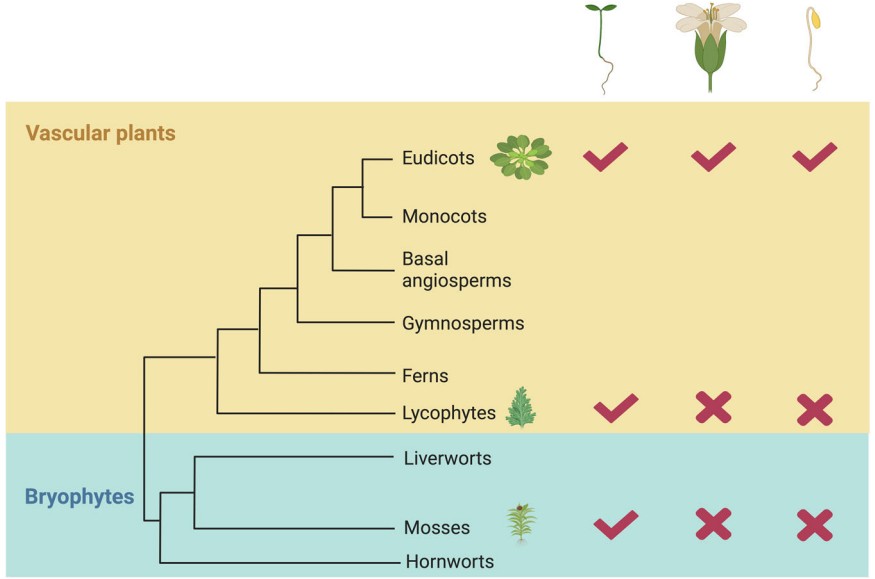

**Fig. 9 Functional divergence of HLS1 in plants.** According to the most up-to-date plant evolutionary relationships[39] and our results, we demonstrated that the functions of HLS1 in the regulation of thermomorphogenesis (elongated seedling cartoon) were conserved from bryophytes to eudicots (shown as check marks). However, the HLS1 orthologs in lycophyte or mosses could not function in flowering time control (flower cartoon) and hook development (depicted as wrong marks). The cartoon was created with BioRender.com.

HLS1 (AtHLS1) complementation lines (Fig. S2a). Ethylene promoted exaggerated apical hook formation, we also tested whether these transgenic lines were responsive to ethylene. Our results showed that even in the presence of ethylene biosynthesis precursor 1-aminocyclopropane-1-carboxylic acid (ACC), these heterologous overexpression lines could not form any hook angles in *hls1-1* mutant backgrounds (Fig. 8c). Therefore, we demonstrated that the HLS1 orthologs from bryophytes and lycophyte were not functional in the regulation of apical hook or flowering time (Fig. 9). That is to say, the function of HLS1 in the

control of cell elongation (thermomorphogenesis) is ancient, but its role in hook development or flowering is obtained in eudicots.

## Discussion
*HLS1* was among the last few genes, which had been cloned before the release of the Arabidopsis genome sequence, but its biochemical functions remain elusive even today. There are currently two models to explain HLS1 activities. The first model proposes that HLS1 directly acetylates histones. Based on protein sequence similarities, HLS1 has been recognized as a putative

acetyltransferase for almost two decades, and the histone H3 acetylation (H3Ac) levels are altered in *hls1* mutants. However, isolated HLS1 protein failed to directly acetylate histones in an in vitro enzymatic assay[9]; thus, the acetyltransferase activity of HLS1 is still under debate. In order to answer whether acetyl-transferase activity is required for HLS1 function, we also generated GFP-HLS1$^{V108A\ L151A}$/*hls1-1* transgenic plants, which harbored two point mutations (V108 and L151) in the conserved acetyltransferase domain (Fig. S5 and Fig. S8). The V108 and L151 amino acids are highly conserved not only in plants but also in bacterial, yeast, and human N-acetyltransferase[3]. All the individual transgenic lines harboring the HLS1$^{V108A\ L151A}$ mutations could not complement the *hls1-1* early-flowering phenotypes, while the normal GFP-HLS1 complementation lines successfully rescue *hls1-1* mutants (Fig. S8). Although we did not directly test the acetyltransferase activity in HLS1, our results suggest that at least these two conserved amino acids in the acetyltransferase domain are required for HLS1 function.

The second model proposes that HLS1 forms oligomers in darkness. Light-activated phytochromes directly interact with HLS1 and reduce its oligomerization status to inhibit hook development[10]. In addition, a very recent study showed that SUMO E3 ligase SAP AND MIZ1 DOMAIN-CONTAINING LIGASE1 (SIZ1) mediates HLS1 SUMOylation and revealed six SUMOylation sites (lysine [K] 62, K81, K155, K186, K294, and K336) in HLS1 protein. Mutation of these SUMOylation sites disturbs the HLS1 oligomerization status and functions in apical hook development[25]. Interestingly, we mapped all these six Arabidopsis HLS1 SUMOylation sites with other orthologs and found that the K62, K294, and K336 sites are completely distinct in bryophytes or lycophyte (Fig. S5). The K81/K155 sites in *S. moellendorffii* are identical with *A. thaliana* HLS1 but not the same in *M. polymorpha* and *P. patens*. The K186 site is not changed in *M. polymorpha* and *P. patens* but is different in *S. moellendorffii* (Fig. S5). Since it has been demonstrated that the oligomerization status of HLS1 is correlated with its role in hook development[10,25], we suspect that the un-conserved SUMOyla-tion sites in HLS1 orthologs might explain why these HLS1 orthologs could not complement the *hls1-1* hook defects.

On another side, the exact role of HLS1 in hook development is unknown. Although it is no doubt that HLS1 is required for apical hook development and asymmetric auxin distributions[4], the signaling mechanisms from HLS1 to the auxin intercellular or intracellular transport are not clear. We only understand that ethylene directly induces *HLS1* expression through the activation of transcription factor EIN3[4,13], but the downstream events from HLS1 are enigmatic.

In the present study, we tried to address these questions through a different angle. We first traced the origin of HLS1 in plants and found that HLS1 originated in embryophytes (Fig. 1). We did not identify any HLS1 orthologs in algae species, suggesting that the functions of HLS1 are likely related to the adaptation of land.

Then we revealed a novel physiological function of HLS1 and uncovered its mechanisms. We demonstrated that HLS1 inter-acted with CO (Fig. 4), associated with the CO binding site in the target promoter (*FT*) (Fig. 5a), and regulated *FT* transcription (Fig. 3d and Fig. 5b). This unexpected signaling mechanism of HLS1 in flowering time control is reminiscent of its role in thermomorphogenesis, which directly interacts with PIF4 and co-regulates a plenty of PIF4-target gene expressions[11]. HLS1 acts as positive regulator in thermomorphogenesis and negative reg-ulator in flowering time control, respectively. Therefore, HLS1 functions as a scaffold protein that associates with multiple transcriptional regulators during different stages of plant growth and development. This unified model is reminiscent of findings

for FRIGIDA (FRI), another protein involved in regulating flowering time in *A. thaliana*. FRI forms a supercomplex with histone acetyltransferases and histone methyltransferases that localizes to the transcriptional regulatory regions of *FLOWERING LOCUS C* (*FLC*)[26]. In future, an in vivo co-immunoprecipitation coupled mass-spectrometry (co-IP/MS) analysis could determine whether the HLS1 protein complex contains other proteins and reveal their identities.

Lastly, we determined the functional divergence of HLS1 from different linages. Interestingly, we found that HLS1 orthologs from *M. polymorpha*, *P. patens,* or *S. moellendorffii* could com-plement the thermomorphogenesis defects in *hls1-1* mutants (Fig. 6), but could not rescue the early-flowering phenotypes and the hookless phenotypes in etiolated seedlings in *hls1-1*. (Figs. 7–8). These results indicate that HLS1 proteins from bryophytes or lycophyte are able to modulate cell elongation phenotypes in *A. thaliana* likely through a conserved gene reg-ulatory network. In fact, it has been reported that cell elongation was also an ancestral ethylene response. In contrast to inhibition of cell elongation in *A. thaliana*, ethylene actually promotes cell elongation in Charophyta[27]. On another side, although we did not find any HLS1 orthologs in charophytes, the counterpart of EIN3 transcription factor in Charophyta (*Spirogyra pratensis*) can partially rescue the *A. thaliana ein3* mutants (*35S-SpEIN3-YFP*/*ein3-1*) and even trigger exaggerated hook formation when overexpressed in Col-0 background (*35S-SpEIN3*/Col-0)[27]. Therefore, the role of EIN3 in charophyte is conserved with *A. thaliana* ortholog, but HLS1 functions more divergently during evolution. We believe that the future knock-out of *HLS1* in the model plant *P. patens* and *M. polymorpha* will tell the role of HLS1 in bryophytes and shed new light on our understanding of this enigmatic protein.

We also discovered that the mutant form of HLS1 in *hls1-1*, which has a single amino acid substitution (HLS1$^{E346K}$), could not interact with transcription factor CO (Fig. 4c). This result indicates that this amino acid (E346) is essential for the function of HLS1 in the regulation of CO activity. However, the E346 site is highly conserved in our analysis (Fig. 1 and Fig. S5) and could not explain why other HLS1 orthologs are not able to comple-ment *hls1-1* early-flowering phenotypes if E346 matters. We assumed that the protein structure or SUMOylation in HLS1 orthologs might affect their interactions with their CO counter-parts in *M. polymorpha*, *P. patens,* or *S. moellendorffii*. It is noteworthy to test their individual HLS1-CO interactions and compare predicted protein structural differences in future.

Taken together, we reported the origin and evolutionary his-tory of HLS1 and pointed out the functional divergence of HLS1 in different plant linages.

## Materials and methods

**HLS1 ortholog identification and phylogenetic analyses**. To obtain orthologs of HLS1 from Archaeplastida, we performed similarity search using genomes from 32 representative plants, comprising 23 streptophytes (land plants and charophytes), 6 chlorophytes, 2 rhodophytes, and 1 glaucophyte (Supplementary Data 1). *Arabi-dopsis* AtHLS1 (At4g37580) was used as template to perform BLASTp searches against theses plant proteomes with low stringency (*E* value < 0.01). We also used BLASTp algorithm to search against the Phytozome v13 (https://phytozome-next.jgi.doe.gov/) and 1KP dataset (1000 plant transcriptomes, https://db.cngb.org/onekp/) to obtain putative HLS1 orthologs from glaucophytes, rhodophytes, chlorophytes, and charophytes as much as possible. We further analyzed the function domains of *Arabidopsis* HLS1 (At4g37580, PF00583, "N-acetyltransfer-ase" domain) for filtering these similar sequences with hits (*E* value < 0.01). The filtered protein sequences were separately aligned using MAFFT (v7)[28], and then trimmed by trimAL (v1.3) with -gt = 0.03[29]. The multiple sequence alignment was manually checked for quality and ensured the completeness of function domains. The maximum likelihood phylogenetic tree was reconstructed using IQ-TREE 2[30]. The best-fitting model was determined by ModelFinder[31], and branch supports were evaluated using the ultrafast bootstrap (UFBoot) approach and SH approx-imate likelihood ratio test (SH-aLRT test) with 1000 replicates[32,33]. The motifs

were predicted by the MEME suit version 5.5.0 (https://meme-suite.org/meme/tools/meme).

**Plant materials and growth conditions**. The *Arabidopsis hls1-1* (point mutation line), *hls1-27* (SALK_136528, T-DNA inserstion line), and *hls1-28* (SM_3_50, transposon insertion line) mutants used in this study were described previously[3,12]. The *co-2* and *ft-10* mutants were obtained from The European Arabidopsis Stock Centre (NASC). The *co-9* mutants were described before[34]. The *hls1-1 co-9, hls1-1 co-2,* and *hls1-1 ft-10* double mutants were generated by genetic crosses between *hls1-1* and *co-2* and between *hls1-1* and *ft-10*, respectively. Homozygous double mutants were verified by PCR-based sequencing in areas flanking the mutated sites. Genotyping primers were listed in Supplementary Data 2. For the complementation test, *35S:MYC-HLS1/hls1-1* plants (overexpression of the MYC tagged HLS1 coding sequence driven by the 35S promoter in *hls1-1* background) have been described[7]. The coding sequences of *MpHLS1, PpHLS1-1, PpHLS1-2,* and *SmHLS1-1,* and *SmHLS1-2* were commercially synthesized (GENEWIZ) and further cloned into pEGAD-LUC vector[35] to obtain LUC-fusion constructs. The mutated AtHLS1 $^{V108A\ L151A}$ fragments were generated by following the instruction manual of the Mut Express MultiS Fast Mutagenesis Kit V2 (Vazyme). The coding sequences of *AtHLS1* and *AtHLS1* $^{V108A\ L151A}$ were cloned into pEGAD-GFP vector[35] to obtain the GFP-fusion constructs. Each construct was then individually transfected into *Agrobacterium tumefaciens* GV3101. *A. tumefaciens* strains harboring each construct were then transformed into *hls1-1* plants through flowering dip method. Obtained transformants were initially screened on Basta containing MS medium (Sigma-Aldrich, #M5519). Basta-resistant lines were picked up and transferred into soil for further immuno-blot confirmation.

To observe thermomorphogenesis phenotypes, seedlings germinated on MS medium were firstly grown under 22 °C (16 h light/8 h dark) for three days and then kept at 22 °C or 28 °C for additional four days. Hypocotyls were imaged under a dissecting microscope (Nikon) and then measured with Image J software (https://imagej.nih.gov/ij/). Statistics were carried out with Prism (GraphPad Software). To examine flowering time, seeds were first placed on MS medium. After stratification for three days, the plates were incubated in a growth chamber for one week. The seedlings were transferred to soil and grown under long day (16 h light/8 h dark, 22 °C) conditions as indicated until flowering. To observe hook angles, seeds were placed on MS medium or MS supplemented with 1 μm of ACC. After stratification for 3 days, the plates were exposed with white light irradiation for three hours and then kept in complete darkness for four days. Hook angles were recorded under a dissect microscope (Nikon).

**Yeast two-hybrid assay**. The coding sequences of *HLS1, CO,* and their truncated forms were PCR amplified, cloned into the pGBKT7 or pGADT7 vector, and transformed into yeast strain AH109 using the Matchmaker Gold Yeast Two-Hybrid System according to the user manual (Clontech, #630489). Transformed yeast cells were streaked onto SD (−Leu/−His/−Ade/−Trp) medium and grown at 28 °C for 4–7 days. The white colonies represented protein–protein interactions. Primers are listed in Supplementary Data 2.

**RNA extraction and qRT-PCR**. Seedlings grown in the light were harvested and ground into powder for RNA extraction. Total RNA was extracted from the samples with TRIzol Reagent (Invitrogen, #15596206). Reverse transcription (Vazyme, #R223) and quantitative PCR were performed according to the manufacturer's instructions (Vazyme, #Q111-02). Expression analysis was performed with three biological replicates. The relative expression levels were normalized against *ACTIN7* as an internal control. Primers used in this study are listed in Supplementary Data 2.

**GUS staining assay**. 12-day-old seedlings grown in the light (LD) were collected at different Zeitgeber times (ZT) and incubated in GUS staining solution at 37 °C for equal amounts of time. After staining, the seedlings were washed with 75% ethanol and observed under a stereoscopic microscope (Nikon) for imaging.

**Chromatin-immunoprecipitation PCR**. Two-gram samples of 10-day-old seedlings grown under LD conditions at ZT-16 were collected and cross-linked in 1% formaldehyde according to the standard ChIP protocol[5]. EZview Red Anti-c-Myc Affinity gel (Sigma-Aldrich, #E6654) was used to immunoprecipitate the DNA that bound to MYC-HLS1. After elution, qPCR was performed to examine interactions. The various regions of the *FT* promoter were previously described[22]. Primers used in this assay are listed in Supplementary Data 2.

**Firefly luciferase complementation imaging assay (LCI)**. The coding sequence of *HLS1* was cloned into pCAMBIA1300-cLUC, while the coding sequence of *CO* was cloned into pCAMBIA1300-nLUC[36]. *Agrobacterium tumefaciens* GV3101 cells harboring the indicated constructs was infiltrated into *Nicotiana benthamiana* leaves using the standard protocol[37]. Luciferase activity was detected with a CCD camera (Tanon, #4100). In each analysis, at least three biological replications were performed with similar results.

**Transient transcriptional activity assay**. The *pFT:LUC* reporter construct was described previously[38]. For the effector constructs, the coding sequence of *CO* was cloned into pCambia1300-HA, while the coding sequence of *HLS1* was cloned into pEGAD to generate GFP-HLS1. *A. tumefaciens* carrying the reporter or effector construct was cultured to OD$_{600}$ = 0.5, and the cultures were combined for infiltration. Luciferase activity was recorded with a CCD camera (Tanon, #4100) or quantitatively measured under a luminometer (Tecan).

**BiFC assays**. The coding sequence of *CO dNT* was cloned into the pEarleyGate201-YN vector, while *HLS1* or *HLS1*$^{E346K}$ was cloned into the pEarleyGate 202-YC vector, using the Gateway system (Invitrogen). The constructs were introduced into *A. tumefaciens* strain GV3101 and co-expressed in *N. benthamiana* leaves. After two days of incubation, the fluorescent signal of yellow fluorescent protein (YFP) was detected under a confocal laser-scanning microscope. The nuclear marker mCherry-VirD2NLS was simultaneously co-expressed for indicating nuclei positions in leaf pavement cells.

**Statistics and reproducibility**. Statistical analyses of data were performed using GraphPad Prism 7.0 software. The hypocotyl length, hypocotyl cell length, and root length were quantified by ImageJ software. Results are shown as the means ± standard deviation (SD). Statistical significance was analyzed by one-way or two-way analysis of variance and post hoc Tukey's test, with different lowercase letters indicating significant differences ($P < 0.05$). Significance levels are: $*P < 0.05$; $**P < 0.01$; $***P < 0.001$.

**Reporting summary**. Further information on research design is available in the Nature Portfolio Reporting Summary linked to this article.

## Data availability

All data that support the findings of this study are available from the corresponding author upon request. Source data for hypocotyl length underlying the graphs are provided in Supplementary Data 3. Accession numbers: these sequences were deposited in GenBank under the following accession numbers: MpHLS1 (ON210989), PpHLS1-1 (ON210990), PpHLS1-2 (ON210991), SmHLS1-1 (ON210992), SmHLS1-2 (ON210993).

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

## Acknowledgements

We thank Dr. Lin Li for the *pFT*:LUC plasmid, Dr. Chentao Lin for pEGAD-LUC vector, Dr. Yongfu Fu for the BiFC vectors, and Dr. Shiyong Song for *co-9* mutants. This study was supported by the National Natural Science Foundation of China (31970256), the Natural Science Foundation of Jiangsu Province (BK20201371), the Key Laboratory of Molecular Design for Plant Cell Factory of Guangdong Higher Education Institutes (2019KSYS006), the QingLan Project of Jiangsu Province and the Priority Academic Program Development of Jiangsu Higher Education Institutions.

## Author contributions

Z.Zhu. designed the research. Q.W. and R.W. generated transgenic plants and performed most of the experiments. N.L. and H.J. characterized flowering phenotypes and initiated yeast two-hybrid assays. J.S. and Z.Zhang. performed evolutionary studies. B.Z. commented on phylogeny and revised manuscript. All authors analyzed data. Z.Zhu. wrote the manuscript with inputs from all authors.

## Competing interests

The authors declare no competing interests.
