## [Peer Review File · Communications Biology]

Reviewers' comments:

Reviewer #1 (Remarks to the Author):

NatComms- 13599_0

The origin, evolution and functional divergence of HOOKLESS1 in plants

The manuscript by Wang et al. describes a novel role for HOOKLESS1 protein regulating flowering time in Arabidopsis. The genetic experiments supporting HSL1 upstream of CO and FT in flowering are clear and solid. The authors also identified HSL1 orthologs of Physcomitrium patens and Selaginella moellendorffii and tested their ability to complement the Arabidopsis mutant phenotypes including flowering time, thermomorphogenesis sensitivity and hook formation finding interesting insights into the evolution of HSL1 function.

Major concerns:

- 1) The authors claim that "HLS1 originates in embryophytes". In any case, if you don't find orthologs in algae genomes/transcriptomes you can't say they don't have it, you can say that HLS1 is detected in angiosperms. Even more, one of the main flaws of your analysis is that the data search for HLS1 orthologs was limited to 32 species, with only 9 algae species. I strongly recommend the authors to increase the data search using other databases such as Phycocosm (genomic, JGI) or 1KP (transcriptomic, CNGB) that has a large number of algae species. This will definitely increase the strength of your analysis to support your claim. In this case, the more data the better.
- 2) How do you search or navigate into data with low sequence similarity due to lineage divergence? Your first sequence search has to be done with low stringency in order to get sequences that resemble your query. The next step is to sort them out into family/subfamily clades using a protein alignment of conserved domains, and finally apply a Maximum Likelihood (ML) analysis or similar to classify the groups. Throughout the analysis you have to use reference sequences to HSL-like sequences following the current nomenclature.
- 3) I disagree in one of the main take home messages of this paper when the authors claim that "These results suggest that during plant evolution, the role of HLS1 in cell elongation (thermomorphogenesis) is ancient, while its role in hook development is co-evolved in angiosperms with other key mechanisms." First, the authors demonstrated that moss and lycophyte HLS1 protein is able to rescue the thermomorphogenesis deficiency of the Arabidopsis mutant plant. In order to claim that HLS1 plays a conserved (not ancient) role in thermomorphogenesis, the moss and lycophyte mutants for HLS1 should develop a thermomorphogenesis phenotype. The authors have not performed these experiments in moss and lycophyte. The authors described here the ability of moss and lycophyte HLS1 to complement the Arabidopsis phenotype, likely through a conserved interactions that support a similar gene regulatory network. So, I suggest to rewrite these claims throughout the manuscript. Secondly, the authors did not quantify cell elongation to be linked to the thermomorphogenesis phenotype. And third, saying that hook development in angiosperms co-evolved with other plant traits is wrong. You meant that hook development is new trait of angiosperms, something that it is not clear to me for the reason below.
- 4) The complementation experiment looks like moss and lycophyte HLS1 are able to partially (not completely) able to restore the hsl1-1 mutant phenotypes, both thermomorphogenesis (Fig. 6) and flowering time (Fig. 7). Why not saying this instead of claiming that is a complete restoration of the phenotypes?
- 5) Since the authors claim that the thermomorphogenesis phenotype is associated to the control of cell elongation, it would be nice to confirm that the thermomorphogenesis phenotype of pEGAD-HSL1 plants is indeed a consequence of a major cell elongation dependent on HSL1.

The main minor concerns:

- 1) mention to early or late land plant species:

-This is a misleading concept that should be corrected throughout the manuscript. Since *P. patens* and *S.a moellendorffii* are extant plants, these plants have evolved as much as angiosperms. I recommend the authors to refer them directly as moss and lycophytes.

2) Please clarify in the main text and in Fig. S2 legend the promoter used to drive the expression of MYC-HLS1 in the *hls1-1* mutant.

3) Error bars in several figures do not state what they represent (SD, SE, CI), such as in Fig. 2B, 3B, 3D.

3) Please re-draw figures using Box-plot elements (e.g. center line, median; box limits, upper and lower quartiles; whiskers, 1.5x interquartile range; points, outliers).

4) Fig 6 B it has no statistic. From the graph, the complementation experiment looks like a partial restoration of the *hsl1-1* mutant phenotype.

5) In the Discussion the authors suggest an interesting experiment that can support the functional conservation of HLS1 orthologs. It would be nice to transform *Arabidopsis hsl1* mutant using an E346 mutant version of PpHLS1-1 and SmHLS1-1. In addition, the authors also suggest an interesting domain swap experiment to be performed on PpHLS1-1 and SmHLS1-1 proteins.

Minor concerns:

-mention to "did not exist in early green plants" should be replace by green algae.

-In Fig. 1 you show the conserved residues L327 and E346 of AtHSL1, that you also found in the bryophyte and fern clades. It is not clear to me if these residues are conserved throughout the angiosperms species and you didn't highlight it for simplicity or if they are not conserved in the rest of the angiosperms HSL1 proteins. Please clarify it in the caption.

- Fig 1 caption: please define the reference to the different colors of highlighted boxes.

-"We obtained one transgenic line for GFP-pHLS1-2 (#9), three lines for GFP-PpHLS1-2 (#2, #5 and #7)" one of them should be GFP-pHLS1-1

-Fig 8C add a reference in the figure to the ethylene treatment. The figure should be self-explicative.

Reviewer #2 (Remarks to the Author):

In this manuscript by Wang et al, the authors show that HLS1 has a role in flowering and hypocotyl elongation besides its well-known role in hook apical development. They went further to show that HLS roles in hook development appeared later in evolution and are not present in early land plants, but hypocotyl and flowering roles are ancient.

The work is interesting for a wide plant biology community, but still preliminary in some of its facets. The data does not convincingly show that HLS requires CO for functioning and evidence that both bind to the same motif is circumstantial. In other words, the same data can also be interpreted as HLS and CO may have largely independent roles on the FT promoter. The fact that HLS1 has a role in flowering is known since the original work by Lehman (Cell, Vol. 85, 183–194, April 19, 1996). Therefore, insights in the mechanistic of HLS1 action is of importance.

More specifically my concerns are:

1) In figure 5 the authors used *co-2 hls1-1* double mutants. The *co-2* allele was originally obtained in the Ler background. Ler and Col have large differences in flowering time. Therefore, the results shown in figure 5B are not of good quality. The assay in Fig S4 should be quantitative. Further, the effects of *hls1-1* in *co* and *ft* backgrounds are still visible. It seems that the effects of *hls1* are additive with those of *co*, rather than epistatic. The authors could use the *co-9* allele they already have, perform quantitative flowering assays, and determine the epistatic relationship between *hls1* and *co-9* mutations, by using appropriate tests and taking care of heteroscedasticity. It would be desirable to quantitate both rosette and cauline leaves that emerged from the apical meristem before flowering (not those appearing from axillary meristems).

2) The transient assay with *N. benthamiana* leaves to test direct regulation of CO and HLS of an FT:LUC reporter may not be significant. Firstly, there is only about a 30% increase in reporter expression after the addition of CO (Fig 5F). This is a subtle effect if compared to what occurs when FT is induced in *Arabidopsis*. Secondly, FT is normally expressed in specialized companion cells (www.pnas.org/cgi/doi/10.1073/pnas.1719455115). Therefore, the relevance of a transient expression in mesophyll cells is unclear.

3) The authors compared the capacity of GFP-PpHLS1-2 and GFP-SmHLS1-1 constructs to complement the *hls1* mutation with that of MYC-HLS1. They concluded that PpHLS1 and SmHLS1 cannot complement the hook phenotype, as MYC-HLS1 did. However, they should have compared the three genes in the same construct background, fused to the same tag. At this point authors cannot rule out that GFP fusion to PpHLS1 and SmHLS1 may disturb some of their properties while the MYC tag may not disturb HLS1.

4) For Co-IP and pull downs, the equivalence of input and precipitated material should be indicated. How much of the protein was precipitated in both assays? Further, MBP alone seems to be pulled down by GST as well.

Minor points

5) The characteristics of complementation constructs should be clearly stated even if the authors requested the constructs from another lab. Is MYC-HLS under 35S or its own promoter? Is it a genomic clone or is it of cDNA origin? It is only mentioned in fig5C legend.

6) Check at the end of page 8. GFP-PhHLS1-2 is repeated

Response to reviewer comments

Dear Editor and Reviewers:

Thank you for reviewing our manuscript. We have made substantial revisions in the past half year according to the suggestions from you and anonymous reviewers. First, we re-performed phylogeny analysis with expanded taxon sampling of algae genomes and improved our searching algorithm according to reviewer suggestions to make our analysis more accurate. Second, we generated new *hls1-1 co-9* homozygous mutants for genetic analysis. Third, we produced new BiFC results to confirm the CO-HLS1 protein-protein interactions and generated new transient promoter activity assays. These results further demonstrate that HLS1 abrogates the CO transcriptional activity. Last but not least, in our previous version, we did not obtain all the five constructs (HLS1 from moss and lycophyte) and did not get homozygous lines for phenotypic comparisons. Here we constructed a series of Luciferase (LUC) tagged transgenic plants and obtained homozygous lines for characterizing the HLS1 functions from different species. We used these new materials to do phenotypic observations and included them in the revised manuscript. We believe that our efforts have improved the quality of this paper to reach the publishing standard of *Communications Biology*.

Reviewers' comments:

Reviewer #1 (Remarks to the Author):

NatComms- 13599_0

The origin, evolution and functional divergence of HOOKLESS1 in plants

The manuscript by Wang et al. describes a novel role for HOOKLESS1 protein regulating flowering time in Arabidopsis. The genetic experiments supporting HSL1 upstream of CO and FT in flowering are clear and solid. The authors also identified HSL1 orthologs of *Physcomitrium patens* and *Selaginella moellendorffii* and tested their ability to complement the Arabidopsis mutant phenotypes including flowering

time, thermomorphogenesis sensitivity and hook formation finding interesting insights into the evolution of HSL1 function.

Mayor concerns:

1) The authors claim that “HLS1 originates in embryophytes”. In any case, if you don’t find orthologs in algae genomes/transcriptomes you can’t say they don’t have it, you can say that HLS1 is detected in angiosperms. Even more, one of the main flaws of your analysis is that the data search for HLS1 orthologs was limited to 32 species, with only 9 algae species. I strongly recommend the authors to increase the data search using other databases such as Phycocosm (genomic, JGI) or 1KP (transcriptomic, CNGB) that has a large number of algae species. This will definitely increase the strength of your analysis to support your claim. In this case, the more data the better.

Response: Thank you for your suggestions. We have re-performed the analyses with expanded taxon sampling of algae genomes (Supplementary table 1) and revised the incorrect statements. In addition to searching against 32 genomes of representative plants, we also used BLASTp algorithm to search against the Phytozome v13 and 1KP dataset as you suggested to obtain putative HLS1 orthologs from algae as much as possible.

2) How do you search or navigate into data with low sequence similarity due to lineage divergence? Your first sequence search has to be done with low stringency in order to get sequences that resemble your query. The next step is to sort them out into family/subfamily clades using a protein alignment of conserved domains, and finally apply a Maximum Likelihood (ML) analysis or similar to classify the groups. Throughout the analysis you have to use reference sequences to HLS-like sequences following the current nomenclature.

Response: Thank you for your professional concerns. We have taken into account the possible low sequence similarity among highly divergent lineages. As you advised, our similarity search in the first step was performed with low stringency (E-value < 0.01). We further used functional domain to filter the sequence with hits, and the filtered

protein sequences were used to perform phylogenetic analyses. Finally, we used the evidence of both the tree topology and conserved motifs to identify the putative HLS orthologs.

3) I disagree in one of the main take home messages of this paper when the authors claim that “These results suggest that during plant evolution, the role of HLS1 in cell elongation (thermomorphogenesis) is ancient, while its role in hook development is co-evolved in angiosperms with other key mechanisms.” First, the authors demonstrated that moss and lycophyte HLS1 protein is able to rescue the thermomorphogenesis deficiency of the Arabidopsis mutant plant. In order to claim that HLS1 plays a conserved (not ancient) role in thermomorphogenesis, the moss and lycophyte mutants for HLS1 should develop a thermomorphogenesis phenotype. The authors have not performed these experiments in moss and lycophyte. The authors described here the ability of moss and lycophyte HLS1 to complement the Arabidopsis phenotype, likely through a conserved interactions that support a similar gene regulatory network. So, I suggest to rewrite these claims throughout the manuscript. Secondly, the authors did not quantify cell elongation to be linked to the thermomorphogenesis phenotype. And third, saying that hook development in angiosperms co-evolved with other plant traits is wrong. You meant that hook development is new trait of angiosperms, something that It is not clear to me for the reason below.

Response: Thank you for your excellent comments. We revised the text according to your suggestions. We found that new complementation lines could rescue the thermomorphogenesis defects in *hsl1-1* mutants (Fig. 6A and Fig. 6C), and we also quantified the hypocotyl cell lengths to prove that the thermomorphogenesis phenotype is associated with the control of cell elongation (Fig. 6B and Fig. 6D).

4) The complementation experiment looks like moss and lycophyte HSL1 are able to partially (not completely) able to restore the *hsl1-1* mutant phenotypes, both thermomorphogenesis (Fig. 6) and flowering time (Fig. 7). Why not saying this instead of claiming that is a complete restoration of the phenotypes?

Response: Thank you for your suggestions. We have generated a series of Luciferase (LUC) tagged transgenic plants to re-observe the thermomorphogenesis and flowering phenotypes, because our previous GFP tagged lines are not homozygous. We actually prepared these two sets transgenic plants (LUC fusion and GFP fusion) simultaneously, but due to the simplicity of LUC based screening, our students paid more attention on collecting homozygous LUC fusion seeds. Our new complementation lines successfully rescued the thermomorphogenesis defects in *hls1-1* mutants (Fig. 6), however, they did not complement *hls1-1* mutants in flowering time control (Fig. 7). Therefore, we conclude that HLS1 orthologs from *M. polymorpha*, *P. patens* or *S. moellendorffii* could only complement the thermomorphogenesis defects in *hls1-1* mutants, but could not rescue the early-flowering phenotypes.

5) Since the authors claim that the thermomorphogenesis phenotype is associated to the control of cell elongation, it would be nice to confirm that the thermomorphogenesis phenotype of pEGAD-HSL1 plants is indeed a consequence of a mayor cell elongation dependent on HSL1.

Response: Thank you for your comments. We have quantified the hypocotyl cell lengths in this revised manuscript to prove that thermomorphogenesis phenotype is associated with cell elongation (Fig. 6B and 6D).

The main minor concerns:

1) mention to early or late land plant species:

-This is a misleading concept that should be corrected throughout the manuscript. Since *P. patens* and *S. moellendorffii* are extant plants, these plants have evolved as much as angiosperms. I recommend the authors to refer them directly as moss and lycophytes.

Response: Thank you for your comments. We have replaced them to “early-branching/diverging” or “moss and lycophytes” to avoid misunderstanding.

2) Please clarify in the main text and in Fig. S2 legend the promoter used to drive the expression of *MYC-HLS1* in the *hls1-1* mutant.

Response: Thank you for your suggestions. We have include detailed information on this described material (promoter information, construct information and citation) in the text and figure.

3) Error bars in several figures do not state what they represent (SD, SE, CI), such as in Fig. 2B, 3B, 3D.

Response: Thanks for your comments. We have added these information accordingly.

3) Please re-draw figures using Box-plot elements (e.g. center line, median; box limits, upper and lower quartiles; whiskers, 1.5x interquartile range; points, outliers).

Response: Thank you for your comments. We have re-prepared figures using the box-plot elements.

4) Fig 6 B it has no statistic. From the graph, the complementation experiment looks like a partial restoration of the *hsl1-1* mutant phenotype.

Response: Thank you for your comments. We have re-observed the thermomorphogenesis phenotype and re-prepared the graph, and came to a conclusion that the complementation lines could complement the thermomorphogenesis defects in *hsl1-1* mutants.

5) In the Discussion the authors suggest an interesting experiment that can support the functional conservation of HLS1 orthologs. It would be nice to transform Arabidopsis *hsl-1* mutant using an E346 mutant version of *PpHLS1-1* and *SmHLS1-1*. In addition, the authors also suggest an interesting domain swap experiment to be performed on *PpHLS1-1* and *SmHLS1-1* proteins.

Response: Thank you for your suggestions. We think that these experiments need much more time than the revision turnaround time and these results could be organized in a separate report.

Minor concerns:

-mention to “did not exist in early green plants” should be replaced by green algae.

Response: Thank you for your suggestions. We made changes accordingly.

-In Fig. 1 you show the conserved residues L327 and E346 of AtHLS1, that you also found in the bryophyte and fern clades. It is not clear to me if these residues are conserved throughout the angiosperms species and you didn't highlight it for simplicity or if they are not conserved in the rest of the angiosperms HLS1 proteins. Please clarify it in the caption.

Response: Thank you for your comments. These two amino acids were highly conserved in all the branches of the phylogeny tree. We re-prepared the Figure 1 and highlighted these conserved residues.

- Fig 1 caption: please define the reference to the different colors of highlighted boxes.

Response: Thank you for your suggestions. We have re-prepared the Figure 1 and shown the functional domain “N-acetyltransferase”.

-“We obtained one transgenic line for GFP-pHLS1-2 (#9), three lines for GFP-PpHLS1-2 (#2, #5 and #7)” one of them should be GFP-pHLS1-1

Response: Sorry for these mistakes. In this revised version, we replaced all the GFP tagged lines with a series of LUC fused homozygous lines. Therefore, these words have been changed accordingly.

-Fig 8C add a reference in the figure to the ethylene treatment. The figure should be self-explicative.

Response: Thank you for pointing this out. We have added ACC information in Fig 8C.

Reviewer #2 (Remarks to the Author):

In this manuscript by Wang et al, the authors show that HLS1 has a role in flowering and hypocotyl elongation besides its well-known role in hook apical development. They

went further to show that HLS roles in hook development appeared later in evolution and are not present in early land plants, but hypocotyl and flowering roles are ancient. The work is interesting for a wide plant biology community, but still preliminary in some of its facets.

The data does not convincingly show that HLS requires CO for functioning and evidence that both bind to the same motif is circumstantial. In other words, the same data can also be interpreted as HLS and CO may have largely independent roles on the FT promoter. The fact that HLS1 has a role in flowering is known since the original work by Lehman (Cell, Vol. 85, 183–194, April 19, 1996). Therefore, insights in the mechanistic of HLS1 action is of importance.

More specifically my concerns are:

1) In figure 5 the authors used *co-2 hls1-1* double mutants. The *co-2* allele was originally obtained in the Ler background. Ler and Col have large differences in flowering time. Therefore, the results shown in figure 5B are not of good quality. The assay in Fig S4 should be quantitative. Further, the effects of *hls1-1* in *co* and *ft* backgrounds are still visible. It seems that the effects of *hls1* are additive with those of *co*, rather than epistatic. The authors could use the *co-9* allele they already have, perform quantitative flowering assays, and determine the epistatic relationship between *hls1* and *co-9* mutations, by using appropriate tests and taking care of heteroscedasticity. It would be desirable to quantitate both rosette and cauline leaves that emerged from the apical meristem before flowering (not those appearing from axillary meristems).

Response: Thank you for your comments. We have generated *hls1-1 co-9* homozygous double mutants for quantitative flowering assays (Fig. S4 and Fig. 7). We have also quantitated both rosette and cauline leaves that emerged from the apical meristem before flowering as you suggested (Fig. 7C).

2) The transient assay with *N. benthamiana* leaves to test direct regulation of CO and HLS of an *FT:LUC* reporter may not be significant. Firstly, there is only about a 30% increase in reporter expression after the addition of CO (Fig 5F). This is a subtle effect if compared to what occurs when FT is induced in Arabidopsis. Secondly, FT is

normally expressed in specialized companion cells (www.pnas.org/cgi/doi/10.1073/pnas.1719455115). Therefore, the relevance of a transient expression in mesophyll cells is unclear.

Response: Thank you for your comments. We have re-performed transient promoter activity assay with a new effector construct (35S:CO-HA). We also showed representative tobacco leaf image with the transient expression of different constructs in the same leaf for comparison. The LUC activity in the quantitative data showed that the expression of *FT* was induced three times after CO expression, which indicated the activation of *FT* transcription by CO (Fig. 5F). We agreed that *FT* is specifically expressed in companion cells, however, here we only detected *FT* promoter activity with this widely-used transient assay protocol. We actually have compared *FT* expression levels in Col-0 and *hls1* mutants with qRT-PCR in Fig. 2

3) The authors compared the capacity of GFP-PpHLS1-2 and GFP-SmHLS1-1 constructs to complement the *hls1* mutation with that of MYC-HLS1. They concluded that PpHLS1 and SmHLS1 cannot complement the hook phenotype, as MYC-HLS1 did. However, they should have compared the three genes in the same construct background, fused to the same tag. At this point authors cannot rule out that GFP fusion to PpHLS1 and SmHLS1 may disturb some of their properties while the MYC tag may not disturb HLS1.

Response: Thank you for your comments. We have re-performed the complementation experiments with a set of lines expressing *MpHLS1*, *PpHLS1-1*, *PpHLS1-2*, *SmHLS1-1* and *SmHLS1-2* fused with the LUC tag in *hls1-1* background, and obtained similar results that the *HLS1* ortholog cannot complement the hook phenotype. We actually prepared LUC fusion and GFP fusion transgenic plants simultaneously, but due to the simplicity of LUC based screening, our students paid more attention on collecting homozygous LUC fusion seeds. Because these complementation lines can complement the *hls1-1* cell elongation defects under high ambient temperature (Fig. 6), and we clearly see their LUC expressions (Fig. S6) under CCD camera, we believe that these LUC fused HLS proteins are functional.

4) For Co-IP and pull downs, the equivalence of input and precipitated material should be indicated. How much of the protein was precipitated in both assays? Further, MBP alone seems to be pulled down by GST as well.

Response: Thank you for your comments. In this revised manuscript, we removed all the CIB1 (in original co-IP and pull down experiments) related contents for the following reasons. First, CIB1 has four homologs, therefore we have to generate *hls1 cib1 cib2 cib3 cib4* quintuple mutants for analysis. It requires very long time for obtaining materials. Second, because we have generated new *hls1 co-9* double mutants and paid more efforts on the transcriptional assay with CO-HLS1, we believe that CO is enough for explaining the HLS1 function in flowering time control. Third, to strengthen the HLS1-CO interaction claim, we re-performed BiFC and co-expressed a mCherry based nuclear marker. Therefore, we think it is fine to remove CIB1 contents without the hurt of topic.

Minor points

5) The characteristics of complementation constructs should be clearly stated even if the authors requested the constructs from another lab. Is MYC-HLS under 35S or its own promoter? Is it a genomic clone or is it of cDNA origin? It is only mentioned in fig5C legend.

Response: Thank you for pointing this out. We have completed the characteristics of complementation constructs and added the promoter information in the Fig. S2.

6) Check at the end of page 8. GFP-PhHLS1-2 is repeated

Response: Thank you for pointing this out. We have replaced the GFP-PpHLS1-1 and GFP-SmHLS1-1 lines with the new complementation lines expressing *MpHLS1*, *PpHLS1-1*, *PpHLS1-2*, *SmHLS1-1* and *SmHLS1-2* fused with the LUC tag into *hls1-1* mutants.

Reviewers' comments:

Reviewer #1 (Remarks to the Author):

The origin, evolution and functional divergence of HOOKLESS1 in plants

The revised manuscript of Wang et al. has corrected one of my major concerns regarding database search parameters and searched genomes/transcriptomes of HOOKLESS1 protein. Again, the genetic experiments supporting HSL1 upstream of CO and FT in flowering are clear and solid. The authors also identified HSL1 orthologs of *Physcomitrium patens*, *Marchantia polymorpha* and *Selaginella moellendorffii* and tested their ability to complement the *Arabidopsis* mutant phenotypes including flowering time, thermomorphogenesis sensitivity and hook formation finding interesting insights into the evolution of HSL1 function.

Major concerns:

Still, I disagree with one central claim in the abstract and discussion: "These results suggest that during plant evolution, the role of HSL1 in cell elongation (thermomorphogenesis) is ancient, while its role in hook development is co-evolved in angiosperms with other key mechanisms." As I mentioned before, the authors demonstrated that liverwort, moss, and lycophyte HSL1 protein is able to rescue the thermomorphogenesis deficiency of the *Arabidopsis* mutant plant. To claim that HSL1 plays a conserved (not ancient) role in thermomorphogenesis, the authors should show a bryophyte mutant of HSL1 showing a thermomorphogenesis phenotype. The main conclusion of the complementation experiment is that bryophyte HSL1 protein is able to modulate the thermomorphogenesis phenotype in *Arabidopsis* likely through a conserved gene regulatory network.

Regarding the complementation assay, the authors should verify the thermomorphogenesis phenotype with other accessory and complementary responses. I suggest the following complementary assays to show that the pathway is active again: 1) measuring thermomorphogenic response genes in the complemented plants, 2) quantifying the protein accumulation of PIF4 or YUCCA in the complemented plants, 3) it was recently shown that thermomorphogenic response includes root elongation. Does complemented lines restore the ability to elongate their roots? (Lee, S., Wang, W. & Huq, E. Spatial regulation of thermomorphogenesis by HY5 and PIF4 in *Arabidopsis*. *Nat Commun* 12, 3656 (2021).)

Minor concerns:

- 1) mention to early or late land plant species:
-there are still some mentions to early-diverging plants. Please, change them.
- 2) rephrase: "and motifs widely presented in land"
- 3) Fig 2 is about a phenotype already characterized in previous papers. Even in the original paper (Lehamn 1996). The current figure 2 should be combined with the complementation experiment using HSL1 from *Arabidopsis* and other plant species.
- 3) Please complete this sentence: Interestingly, two amino acids (L327 and E346), whose mutation results in hookless phenotype... IN ARABIDOPSIS...
- 4) Please, rephrase this sentence for clarity reasons: "Then we examined three HSL1-involved representative phenotypes in these transgenic plants."
- 5) Add a reference to this sentence "In *hls1-1* mutants, their hypocotyls could not elongate under high temperature due to the defects in hypocotyl cell elongation under high temperature".
- 6) For simplicity, Fig 8B should include the results of *hls1* complementation using the *AtHSL1* genes shown in Fig S2A

Reviewer #2 (Remarks to the Author):

This is a revised version of a manuscript I revised before. The authors have improved their manuscript after first revision but I still have concerns respect the interpretation of some results and the controls missing in one experiment.

1) The authors state that "The homozygous line (#21) of *hls1-1 co-9* double mutants (genotyped in Fig. S4) displayed late flowering phenotypes compared with the *hls1-1* parental lines (Fig. 5A-C), similar as the wild type plants *Col-0*. These results demonstrate that *HLS1* acts genetically upstream of *CO* in the regulation of flowering time". This interpretation is incorrect. The double mutant *hls1 co* flowers much earlier than the *co* single mutant. This implies that *HLS* can delay flowering largely independently of *CO*. Therefore, these data do not support the proposition that *HLS* regulates flowering upstream of *CO*, although it is still possible that some of the flowering effects of *HLS* occur through the interaction with *CO*. The manuscript text should be modified to correct this interpretation issue.

2) The authors state that "The high expression levels of *FT* in *hls1-1* mutants were largely repressed by the loss of *CO* in the *hls1-1 co-9* double mutants (Fig. 5D). These results indicated that *HLS1* functions with *CO* in a common genetic pathway to control the expression of *FT*." Indeed the results might be interpreted otherwise. The expression of *FT* is much higher in the *hls1 co* mutant compared to the *co* mutant alone. The data might be interpreted again as *HLS1* having an important effect on *FT* expression in a parallel pathway to *CO*

3) The authors state that *HLS1* occupies the same binding sites in the *FT* promoter as *CO* to repress *FT* expression. They do not show evidence that this is indeed occurring. Both proteins may be close in the promoter and interact, but they do not show evidence than *HLS1* depends on *CO* to bind the *FT* promoter.

All these three points might be addressed with changes in the text to avoid statements not supported by data. The authors may find useful to change the order of experiments in figure 5, to move the epistasis experiments to the end of the section to test if the protein interactions between *CO* and *HLS1* shown in fig 4 and fig5 imply that *HLS1* depends con *CO* or not (and your data says it doesn't depend totally on *CO*)

4) The complementation constructs shown in fig 6,7,8 lack an essential control. They should have included the *LUC:HLS1* fusion to test if this is capable of complementing all the mutant phenotypes. The *MYC:HLS1* fusion is not comparable, as both tags are different. The large differences in the size of the tags (or even in their sequence) might end in different behavior. For example, the *LUC* tag is huge and could interfere with some interactors explaining the lack of complementation of some of the assays.

Unfortunately, solving this issue needs new plant transformants, but this issue was raised in the first round of revision.

Response to reviewer comments

Reviewers' comments:

Reviewer #1 (Remarks to the Author):

The origin, evolution and functional divergence of HOOKLESS1 in plants

The revised manuscript of Wang et al. has corrected one of my major concerns regarding database search parameters and searched genomes/transcriptomes of HOOKLESS1 protein. Again, the genetic experiments supporting HSL1 upstream of CO and FT in flowering are clear and solid. The authors also identified HSL1 orthologs of *Physcomitrium patens*, *Marchantia polymorpha* and *Selaginella moellendorffii* and tested their ability to complement the *Arabidopsis* mutant phenotypes including flowering time, thermomorphogenesis sensitivity and hook formation finding interesting insights into the evolution of HSL1 function.

Response: Thank you for your positive response to our previous revision.

Major concerns:

Still, I disagree with one central claim in the abstract and discussion: “These results suggest that during plant evolution, the role of HSL1 in cell elongation (thermomorphogenesis) is ancient, while its role in hook development is co-evolved in angiosperms with other key mechanisms.” As I mentioned before, the authors demonstrated that liverwort, moss, and lycophyte HSL1 protein is able to rescue the thermomorphogenesis deficiency of the *Arabidopsis* mutant plant. To claim that HSL1 plays a conserved (not ancient) role in thermomorphogenesis, the authors should show a bryophyte mutant of HSL1 showing a thermomorphogenesis phenotype. The main conclusion of the complementation experiment is that bryophyte HSL1 protein is able to modulate the thermomorphogenesis phenotype in *Arabidopsis* likely through a conserved gene regulatory network.

Response: Thank you for your comments. We rephrased these claims as you suggested in this revised manuscript.

Regarding the complementation assay, the authors should verify the thermomorphogenesis phenotype with other accessory and complementary responses. I suggest the following complementary assays to show that the pathway is active again: 1) measuring thermomorphogenic response genes in the complemented plants, 2) quantifying the protein accumulation of PIF4 or YUCCA in the complemented plants, 3) it was recently shown that thermomorphogenic response includes root elongation. Does complemented lines restore the ability to elongate their roots? (Lee, S., Wang, W. & Huq, E. Spatial regulation of thermomorphogenesis by HY5 and PIF4 in Arabidopsis. Nat Commun 12, 3656 (2021).)

Response: Thank you for your comments. We have detected *YUCCA8* expression levels in wild-type, *hls1-1* mutants and our complementation lines (Figure 6E) to further strengthen our conclusions. We also checked the root elongation phenotypes in these lines. In fact, even *hls1-1* mutants did not display any defects in high temperature induced root elongation (aka root thermomorphogenesis) (Figure S7). It makes sense because our previous publications have demonstrated that HLS1 acts in a PIF4 dependent manner (Jin et al., Plant Communications 2020), while PIF4 is not involved in root thermomorphogenesis according to several reports (Lee et al, Nature Communications 2021; Borniego et al, New Phytologist 2022).

Minor concerns:

1) mention to early or late land plant species:

-there are still some mentions to early-diverging plants. Please, change them.

Response: Thank you for your comments. We changed them accordingly.

2) rephrase: “and motifs widely presented in land”

Response: Thank you for your comments. We rephrased it in this revised manuscript.

3) Fig 2 is about a phenotype already characterized in previous papers. Even in the original paper (Lehamn 1996). The current figure 2 should be combined with the complementation experiment using HLS1 from Arabidopsis and other plant species.

Response: Thank you for your suggestions. Although the original 1996 Cell paper mentioned the *hls1* early flowering phenotype, we now included more additional mutant alleles to strengthen the conclusion. We also added a new figure (Fig. S8) in this revised manuscript. In the Fig. S8, we not only showed AtHLS1 complementation phenotypes (GFP-HLS1/*hls1-1*) but also included GFP-HLS1^{V108A L151A}/*hls1-1* point mutations. These two amino acids are conserved in the acetyltransferase domain in plants, yeast, human and even bacteria. Our data revealed that these two sites are necessary for HLS1 function in flowering time control. To follow the logic, we decide to keep the Fig. 2 in this version but not combine them together.

3) Please complete this sentence: Interestingly, two amino acids (L327 and E346), whose mutation results in hookless phenotype... IN ARABIDOPSIS...

Response: Thank you for your comments. We corrected it in this version.

4) Please, rephrase this sentence for clarity reasons: “Then we examined three HLS1-involved representative phenotypes in these transgenic plants.”

Response: Thank you for your comments. We rephrased it in this revised manuscript.

5) Add a reference to this sentence “In *hls1-1* mutants, their hypocotyls could not elongate under high temperature due to the defects in hypocotyl cell elongation under high temperature”.

Response: Thank you for your comments. We added its reference.

6) For simplicity, Fig 8B should include the results of *hls1* complementation using the AtHLS1 genes shown in Fig S2A

Response: Thank you for your comments. Because our complementation lines shown in Fig. 8 are LUC fusions, we do not think it is adequate to compare them with MYC tagged lines. But since the LUC fused HLS1 orthologs function in thermomorphogenesis, we believe that it is fine to conclude that these LUC fusion proteins are functional. The defects to rescue hook phenotype are not caused by tag or construct but related to their protein physiological functions.

Reviewer #2 (Remarks to the Author):

This is a revised version of a manuscript I revised before. The authors have improved their manuscript after first revision but I still have concerns respect the interpretation of some results and the controls missing in one experiment.

1) The authors state that “The homozygous line (#21) of hls1-1 co-9 double mutants (genotyped in Fig. S4) displayed late flowering phenotypes compared with the hls1-1 parental lines (Fig. 5A-C), similar as the wild type plants Col-0. These results demonstrate that HLS1 acts genetically upstream of CO in the regulation of flowering time”. This interpretation is incorrect. The double mutant hls1 co flowers much earlier than the co single mutant. This implies that HLS can delay flowering largely independently of CO. Therefore, these data do not support the proposition that HLS regulates flowering upstream of CO, although it is still possible that some of the flowering effects of HLS occur through the interaction with CO. The manuscript text should be modified to correct this interpretation issue.

Response: Thank you for your comments and suggestions. We revised our claims to weaken our statement on this issue.

2) The authors state that “The high expression levels of FT in hls1-1 mutants were largely repressed by the loss of CO in the hls1-1 co-9 double mutants (Fig. 5D). These results indicated that HLS1 functions with CO in a common genetic pathway to control the expression of FT.” Indeed the results might be interpreted otherwise. The expression of FT is much higher in the hls1 co mutant compared to the co mutant alone. The data

might be interpreted again as HLS1 having an important effect on FT expression in a parallel pathway to CO

Response: Thank you for your comments and suggestions. We revised our claims to weaken our statement on this issue.

3) The authors state that HLS1 occupies the same binding sites in the FT promoter as CO to repress FT expression. They do not show evidence that this is indeed occurring. Both proteins may be close in the promoter and interact, but they do not show evidence that HLS1 depends on CO to bind the FT promoter.

All these three points might be addressed with changes in the text to avoid statements not supported by data. The authors may find useful to change the order of experiments in figure 5, to move the epistasis experiments to the end of the section to test if the protein interactions between CO and HLS1 shown in fig 4 and fig5 imply that HLS1 depends on CO or not (and your data says it doesn't depend totally on CO)

Response: Thank you for your comments and suggestions. We have changed the order in Figure 5 to clarify the genetic relationship between *HLS1* and *CO* in this revised manuscript according to your suggestions. We also weakened our statements to interpret our results more accurately.

4) The complementation constructs shown in fig 6,7,8 lack an essential control. They should have included the LUC:HLS1 fusion to test if this is capable of complementing all the mutant phenotypes. The MYC:HLS1 fusion is not comparable, as both tags are different. The large differences in the size of the tags (or even in their sequence) might end in different behavior. For example, the LUC tag is huge and could interfere with some interactors explaining the lack of complementation of some of the assays.

Unfortunately, solving this issue needs new plant transformants, but this issue was raised in the first round of revision.

Response: Thank you for your comments. We really appreciate your suggestion on this issue and believe that providing a LUC-AtHLS1/*hls1-1* line will definitely strengthen our conclusion. However, as you mentioned, it requires much longer time and effort to

make it. Nonetheless, we tried to fix this issue with other strategies. First, our complementation lines successfully rescue the thermomorphogenesis defects in *hls1-1* mutants. We also detected *YUCCA8* expression as Reviewer 1 requested in this version to further prove our conclusion (Fig. 6E). This result suggests that all the LUC-HLS1 proteins in the complementation lines are functional. Secondly, we compared two GFP tagged lines, which expressed GFP-HLS1 or GFP-HLS1^{V108A L151A} in *hls1-1* mutant, respectively (Fig. S8). We showed that these two lines display different flowering time phenotypes. These two amino acids are conserved in the acetyltransferase domain in plants, yeast, human and even bacteria. Our data revealed that these two sites are necessary for HLS1 function in flowering time control. Given the different flowering phenotypes in the GFP-HLS1^{V108A L151A}/*hls1-1* and GFP-HLS1/*hls1-1* transgenic plants are due to the distinct functions of their GFP fusion proteins, we believe that the different rescue phenotypes are not brought by LUC tags. Thirdly, we had generated LUC-CRY2 in *cry1 cry2* background (LUC-CRY2/*cry1 cry2*) more than ten years ago and found these materials successfully rescued the *cry1 cry2* late flowering phenotype as the commonly used GFP-CRY2/*cry1 cry2* lines (both were driven by the same 35S promoter). Since the comparable molecular weight between LUC and GFP, we do not think it is likely that LUC has some negative effects on fusion proteins *in planta*. Therefore we usually generate transgenic plants with LUC fusions due to their simplicity on T1 screening. We wish that you can satisfy with our revision.

REVIEWERS' COMMENTS:

Reviewer #1 (Remarks to the Author):

COMMSBIO-22-2082B

This is the third version of the manuscript by Wang et al. The manuscript has substantially improved and has agreed with most of my suggestions.

Minor concerns:

This sentence should be wrong: Why do you say they are "nonallelic hls1 mutants (hls1-27 and hls1-28)". If they're in the same locus, they are allelic mutations. If they are indeed nonallelic mutations, please state where is the mutation and what are they useful for in this story? A little confusing.

I still have some concerns from figure 6. The picture of the seedling phenotypes depicted in Fig 6A do not show a clear complementation compared to the WT. In addition, the SD of some bars appeared so large that makes it difficult to understand how they pass are significant under the current statistical analysis (Fig 6C). Please, add new pictures and show the statistical analysis as a supplementary table including the assumptions and statistical tests used on the data.

In the sentence "These results illustrate that HLS1 postpones the initiation of flowering in plants", I would rather use "inhibits" instead of "postpones".

REVIEWERS' COMMENTS:

Reviewer #1 (Remarks to the Author):

COMMSBIO-22-2082B

This is the third version of the manuscript by Wang et al. The manuscript has substantially improved and has agreed with most of my suggestions.

Minor concerns:

This sentence should be wrong: Why do you say they are “nonallelic *hls1* mutants (*hls1-27* and *hls1-28*)”. If they’re in the same locus, they are allelic mutations. If they are indeed nonallelic mutations, please state where is the mutation and what are they useful for in this story? A little confusing.

Response: Thanks for your comments. These two alleles (*hls1-27* and *hls1-28*) are different mutant alleles, which have been described in our previous publication (Jin et al., *Science China Life Science*, 2019). *hls1-27* is a T-DNA insertion line (SALK_136528), while *hls1-28* is a transposon insertion line (SM_3_50). We included these detailed information in the Materials and Method part in this revised version.

I still have some concerns from figure 6. The picture of the seedling phenotypes depicted in Fig 6A do not show a clear complementation compared to the WT. In addition, the SD of some bars appeared so large that makes it difficult to understand how they pass are significant under the current statistical analysis (Fig 6C). Please, add new pictures and show the statistical analysis as a supplementary table including the assumptions and statistical tests used on the data.

Response: Thanks for your comments. We changed the representative pictures and listed our source data in the Supplementary Table 2 in this revised version.

In the sentence “These results illustrate that HLS1 postpones the initiation of flowering in plants”, I would rather use “inhibits” instead of “postpones”.

Response: Thanks for your suggestion. We made changes accordingly.